https://doi.org/10.1038/s42003-024-05840-3　　**OPEN**
# Genomic attributes of airway commensal bacteria and mucosa

Leah Cuthbertson[1,21], Ulrike Löber [2,3,4,5,21], Jonathan S. Ish-Horowicz[1,6,21], Claire N. McBrien[1],
Colin Churchward[1], Jeremy C. Parker[1], Michael T. Olanipekun [1], Conor Burke[7], Aisling McGowan[7],
Gwyneth A. Davies[8,9], Keir E. Lewis[9,10], Julian M. Hopkin[9], Kian Fan Chung [1], Orla O'Carroll[7], John Faul[7],
Joy Creaser-Thomas[9], Mark Andrews[10], Robin Ghosal[10], Stefan Piatek [1], Saffron A. G. Willis-Owen[1],
Theda U. P. Bartolomaeus[2,3,4,5], Till Birkner [2,3,5], Sarah Dwyer[1], Nitin Kumar [11], Elena M. Turek[1],
A. William Musk[12,13,14], Jennie Hui[12,13], Michael Hunter [12,13], Alan James[12,14,15],
Marc-Emmanuel Dumas [1,16,17,18], Sarah Filippi[6], Michael J. Cox [19], Trevor D. Lawley [11],
Sofia K. Forslund [2,3,4,5,20✉], Miriam F. Moffatt [1,22✉] & William. O. C. Cookson [1,22✉]

Microbial communities at the airway mucosal barrier are conserved and highly ordered, in likelihood reflecting co-evolution with human host factors. Freed of selection to digest nutrients, the airway microbiome underpins cognate management of mucosal immunity and pathogen resistance. We show here the initial results of systematic culture and whole-genome sequencing of the thoracic airway bacteria, identifying 52 novel species amongst 126 organisms that constitute 75% of commensals typically present in heathy individuals. Clinically relevant genes encode antimicrobial synthesis, adhesion and biofilm formation, immune modulation, iron utilisation, nitrous oxide (NO) metabolism and sphingolipid signalling. Using whole-genome content we identify dysbiotic features that may influence asthma and chronic obstructive pulmonary disease. We match isolate gene content to transcripts and metabolites expressed late in airway epithelial differentiation, identifying pathways to sustain host interactions with microbiota. Our results provide a systematic basis for decrypting interactions between commensals, pathogens, and mucosa in lung diseases of global significance.

[1] National Heart and Lung Institute, Imperial College London, London, UK. [2] Max Delbrück Center for Molecular Medicine (MDC), 13125 Berlin, Germany. [3] Experimental and Clinical Research Center, A Cooperation of Charité-Universitätsmedizin Berlin and Max Delbrück Center for Molecular Medicine, Lindenberger Weg 80, 13125 Berlin, Germany. [4] DZHK (German Centre for Cardiovascular Research), Partner Site, 10785 Berlin, Germany. [5] Charité-Universitätsmedizin Berlin, Corporate Member of Freie Universität Berlin and Humboldt-Universität zu Berlin, 10117 Berlin, Germany. [6] Department of Mathematics, Imperial College London, London, UK. [7] Department of Respiratory Medicine, Connolly Hospital, Dublin, Ireland. [8] Population Data Science and Health Data Research UK BREATHE Hub, Swansea University Medical School, Swansea University, Swansea, UK. [9] College of Medicine, Institute of Life Science, Swansea University, Swansea, UK. [10] Respiratory Medicine, Hywel Dda University Health Board, Llanelli, UK. [11] Host-Microbiota Interactions Laboratory, Wellcome Sanger Institute, Wellcome Genome Campus, Hinxton, UK. [12] School of Population and Global Health, The University of Western Australia, Perth, WA, Australia. [13] Busselton Population Medical Research Institute, Sir Charles Gairdner Hospital, Perth, WA, Australia. [14] Department of Respiratory Medicine Sir Charles Gairdner Hospital, Perth, WA, Australia. [15] Department of Pulmonary Physiology and Sleep Medicine, Sir Charles Gairdner Hospital, Perth, WA, Australia. [16] Department of Metabolism, Digestion and Reproduction, Imperial College London, London, UK. [17] U1283 INSERM / UMR8199 CNRS, Institut Pasteur de Lille, Lille University Hospital, European Genomic Institute for Diabetes, University of Lille, Lille, France. [18] McGill Genome Centre, McGill University, Montréal, QC, Canada. [19] University of Birmingham College of Medical and Dental Sciences, 150183, Institute of Microbiology and Infection, Birmingham, UK. [20] Structural and Computational Biology Unit, European Molecular Biology Laboratory, Structural and Computational Biology Unit, 69117 Heidelberg, Germany. [21]These authors contributed equally: Leah Cuthbertson, Ulrike Löber, Jonathan S. Ish-Horowicz. [22]These authors jointly supervised this work: Miriam F. Moffatt, William O.C. Cookson. ✉email: sofia.forslund@mdc-berlin.de; m.moffatt@imperial.ac.uk; w.cookson@imperial.ac.uk

The mucosal surfaces of the airways and lungs are extensive and constantly challenged by inhaled microorganisms[1–3]. Overt respiratory infections are the leading cause of death in developing countries, resulting in 4 million lost lives annually[4]. Asthma and COPD each affect more than 300 million people worldwide and acute exacerbations of both diseases are driven by respiratory infections[5]. Two-thirds of individuals exposed to COVID-19 in their home[6] and half of subjects directly challenged with COVID-19[7] do not develop infections because of unknown resistance factors.

Upper and lower airways contain a characteristic microbiome[8] that is essential to respiratory health[9]. The commensal microbiota regulates immunity in the respiratory mucosa through multiple mechanisms[10–12] that appear within the first days of life[13].

The nose, oropharynx, and the intrathoracic airways form a contiguous tract. The nasopharyngeal mucosa differs histologically and functionally from lower sites[14], as does its resident microbiota[15]. Common pulmonary diseases including asthma, COPD, bronchopneumonia, cystic fibrosis and lung cancer arise in the intrathoracic airways, whose commensal microbiota are similar to those of the oropharynx[8,16,17]. Up and downward microbial movement occurs between sites[17]. Respiratory pathobionts such as *Streptococcus pneumoniae*, *Haemophilus influenzae*, and *Neisseria meningitidis* are commonly carried in the nose and throat without symptoms. The oro-pharyngeal microbiota does not vary greatly between individuals and is organised into co-abundance networks that may share similar niches[18]. Microbial community dysbiosis with overgrowth of pathobionts accompanies asthma, COPD, pneumonia, and other pulmonary disorders[9,19].

The airway microbiota encompasses viruses, fungi, and bacteria[20]. A variable viral microbiome (excluding phage) is well described at the molecular level[20,21]. Oro-pharyngeal fungi such as Candida and Aspergillus spp. are commonly cultured from asthmatics, confounded by therapy with inhaled corticosteroids. Although important in cystic fibrosis and bronchiectasis[22], fungi have very low biomass in the lower airways of healthy individuals[23]. Airway commensal bacteria from healthy subjects have not previously been systematically cultured or sequenced. This lack has limited the structured study of interactions between bacteria, viruses, fungi, and mucosal immunity in clinical samples or in model systems. In this paper we describe such systematic exploration, substantially extending what is known about core constituents of airway bacterial communities.

Our study design is summarised in Supplementary Fig. 1. We have used mucin-enriched media to culture and sequence novel taxa that account for 75% of the abundance of airway commensal bacteria. Functional characterisation, evolutionary analyses, and comparison with amplicon sequencing in representative human samples extend the scope of these results.

## Results

**Culture collection and isolate novelty.** Lower airway bacteria were cultivated from bronchoscopic brushings from two asthmatics and three healthy individuals from the Celtic Fire Study (described below). We used a limited range of media with and without 0.5% mucin, followed by incubation in a standard atmosphere or an anaerobic workstation to capture 706 isolates. Those without overlapping 16S rRNA gene sequences were transferred to the Wellcome Sanger Institute and the whole-genome sequenced with assembly using Bactopia (v 1.4.11).

We cultured 651 isolates, 256 of which were successfully whole-genome sequenced. Of these, five sequences appeared mixed and were excluded. After removing duplicates on a 99.5% nucleotide identity threshold, 126 unique strains remained. The

Bactopia quality report for the genome assemblies is reported in Supplementary Data 1. Forty-four isolates were annotated to species level in accordance with MIGA[24] (TypeMat and NCBIProk) and with GTDBtk. A further 30 species were identified by either MIGA (TypeMat and NCBIProk) or GTDBtk. The genome completeness and the contamination percentage were tested within the MIGA pipeline aligning 106 bacterial core genes[25] (Supplementary Data 2).

All isolates were assigned to genera in the TypeMat or NCBI prokaryotes database with $P < 0.05$. Among these samples, we classified 49 *Streptococcus*, ten *Veillonella*, nine each of *Gemella* and *Rothia*, eight *Prevotella*, six each of *Neisseria*, *Micrococcus* and *Pauljensenia*, five each of *Haemophilus* and *Staphylococcus*, three *Granulicatella*, two each of *Actinomyces*, *Cutibarterium* and *Fusobacterium* and one *Cuprividis*, *Leptotrichia*, *Microbacterium* and *Niallia*, respectively (Fig. 1a).

We defined a 'new species' when isolates could not be assigned to known species in reference databases[24]. We classified isolates as 'putatively novel species' when they exhibited no close relation to any species in the TypeMat or NCBI Prokaryotic Databases, determined by the MIGA tool with a $P$-value threshold of 0.05 and an incongruent species assignment indicated by gtdbtk.

Fifty-two isolates could not be assigned with $P < 0.05$ to known species in the reference databases[24] (Fig. 1b). Twenty-eight of the putative novel species were contained within the *Streptococcus* genus, six within *Pauljensenia* (not previously recognised to be prevalent in the airways), and four each within *Neisseria* and *Gemella* (Fig. 1c and Supplementary Data 1).

Comparison of the entire sequences of our streptococcal isolates with 2477 public *Streptococcus* spp. sequences showed that the organisms were widely distributed amongst *S. infantis*, *S. oralis*, *S. mitis*, *S. pseudopneumoniae*, *S. sanguinis*, *S. parasanguinis*, and *S. salivarius* (Supplementary Fig. 2).

### Isolate characteristics

*Kegg Orthology of isolate genomes.* We used the eggNOG (evolutionary genealogy of genes, Non-supervised Orthologous Groups) mapper tool (as previously for large-scale systematic genome annotations[26]) to assign by transfer 5,531 Kegg Ontology (KO) annotations for the 126 isolates. We encoded these in a binary matrix indicating presence or absence (Supplementary Data 3) and constructed an isolate phylogeny after removing 254 zero-variance KOs (either present or absent in all isolates) and reducing identical KO presence/absence to single examples before hierarchical clustering with the Manhattan distance metric and complete linkage. The Dynamic Tree Cut algorithm[27] identified 15 clusters of isolates that recovered known phylogenetic relationships (Fig. 2a). Based on the observed 16 S rRNA gene sequence similarity, we further divided one *Streptococcus* cluster into two (Strep I and Strep II, Fig. 2a). Relative KO enrichment was estimated for each of the 16 clusters by contingency table analysis.

Annotation for the 5277 informative KOs (including duplicates removed during clustering) (Supplementary Data 4) identified 247 uncharacterised proteins (Supplementary Data 4). Features of particular interest among the known genes are summarised below.

*Biofilms.* Biofilm formation is a feature of respiratory pathogens, archetypically *Pseudomonas* spp. in patients with cystic fibrosis. Biofilm-associated genes were also common in the commensal collection (Supplementary File 4b). Ninety genes were annotated with "biofilm" in their KO pathway descriptions, with *cysE* (serine O-acetyltransferase), *vpsU* (tyrosine-protein phosphatase), *luxS* (S-ribosylhomocysteine lyase), *trpE* (anthranilate synthase

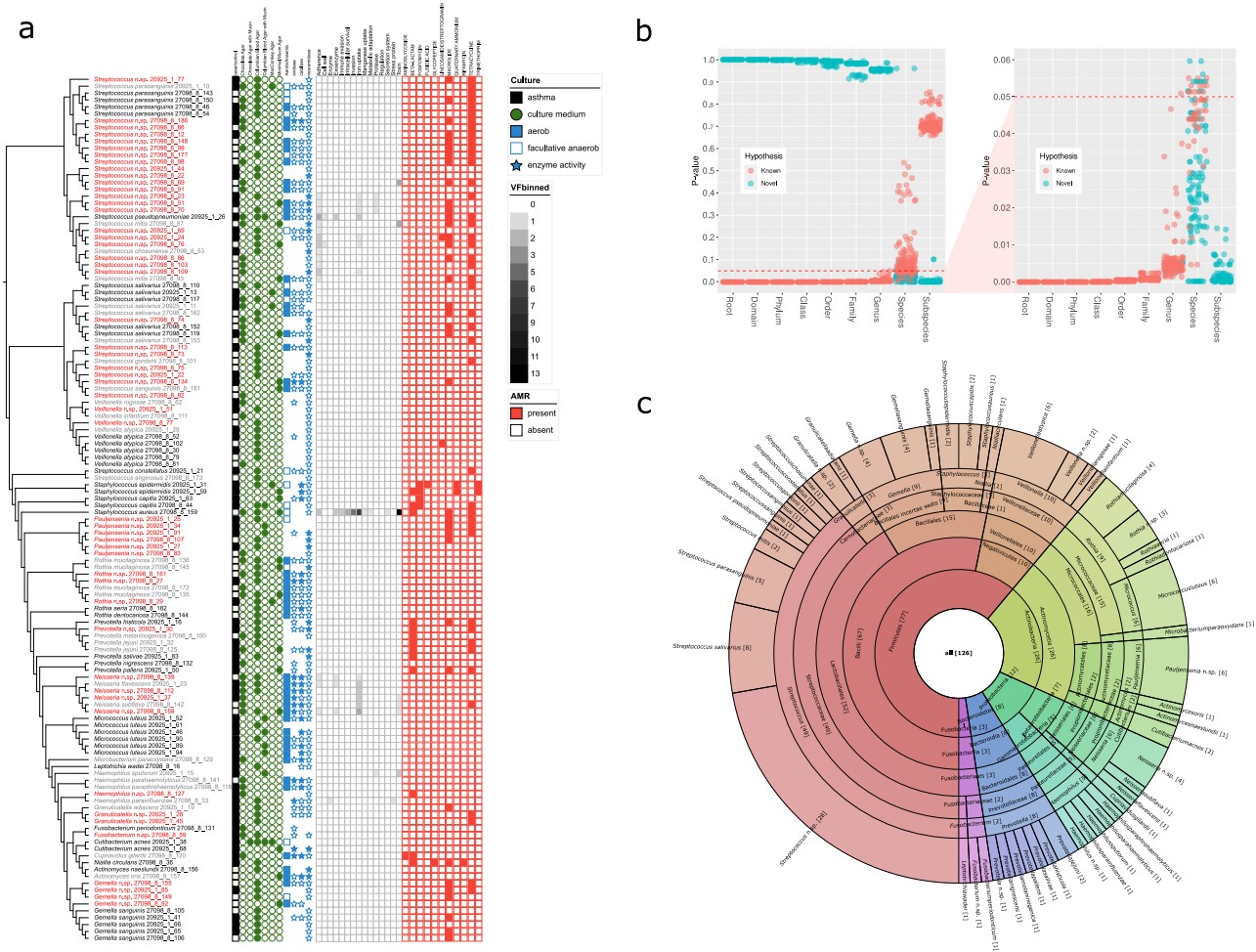

**Fig. 1 Genomic characteristics of airway mucosal bacteria. a** Culture collection phylogeny based on average nucleotide identities between genomes with 1000 bp fragment length. Putatively novel species are highlighted in red (indicating that it is not related to any species in the TypeMat DB or NCBI Prok DB ($P < 0.05$) when assessed using MIGA and not assigned to a known species or incongruent species assignment using gtdbtk). Greyed-out isolates are not fully supported by MIGA and gtdbtk. Genome completeness and contamination are displayed as a bar chart. AMR finder was used to identify antimicrobial resistance genes at the protein level (red panel). Virulence factors were identified using the VFDB and Ariba databases and binned into 15 categories (heatmap). The asthma status of the host is indicated in the black asthma/control panel. Cultivation conditions are indicated in green circles for selected growth media, blue rectangles for aerobic, and white rectangles for anaerobic cultivation. Positive Gram staining for GNB, GNC, GPB, GPC, and other Gram staining is shown in black circles. The neuraminidase activity was tested if a blue star was present and was filled for the positive test and white for a negative test. **b** Taxonomic novelty as calculated by MIGA using TypeMat reference. The scatterplot shows support (*P*-value, vertical axis) for each taxon relative to complementary hypotheses that this taxon is a previously known one (red markers) or a novel one (cyan markers) at each taxonomic level (horizontal axis). Many of the isolate collections constitute novel species within known genera. **c** Composition of bacteria isolated and cultivated from five subjects. Counts are shown for all lineages from species level (outer circle) to phylum level (inner circle) in squared brackets. The ETE3 toolkit was used to fetch taxonomic lineages for all genera of cultured isolates[101]. The number of unique species was summed up and visualised along with their lineages using Krona tools[102].

component I) and *PYG* (glycogen phosphorylase) present in >75% of isolates. Amongst the most abundant organisms, *Haemophilus* and *Prevotella spp.* had distinctive profiles of other biofilm pathway genes (Supplementary Data 4).

*Antimicrobial resistance and virulence.* Many of our isolates contained known genes for antimicrobial resistance (AMR) against tetracyclines and macrolides. *Staphylococcus*, *Prevotella* and *Haemophilus* spp. also possessed beta-lactam resistance (Fig. 1a and Supplementary Data 4). Virulence factors and toxins were concentrated in *Streptococcus*, *Staphylococcus*, *Haemophilus*, and *Neisseria* spp. (Fig. 1a and Supplementary Data 4). Although these annotations neither guarantee that the genes in question are expressed nor that they drive clinically relevant AMR or virulence, they do indicate such potential.

*Antibiotic and toxin synthesis.* Competition between bacteria is fundamental to maintaining stable communities[28]. Genes with a KO pathway annotation for antibiotic synthesis ($n = 33$) were present in many genera (Supplementary Data 4). Arachin biosynthetic genes included *acpP* (acyl carrier protein) which was present in 120 isolates and *auaG* in seven (mostly *Staphylococcus* spp); *rifB* (rifamycin polyketide synthase) present in 20 (*Veillonella* and *Staphylococcus* spp.); *BacF* (bacilysin biosynthesis transaminase) present in 12 (*Staphylococcus* and *Gemella* spp.); and *sgcE5* (enediyne biosynthesis protein E5) present in 12, mostly *Haemophilus* spp. Bacteriocin exporter genes *blpB* and *blpA* were present in 35 and 31 isolates respectively, predominantly *Streptococcus* and *Pauljensenia* spp. (Supplementary Data 4).

Toxins and antitoxin genes were common in the collection (Supplementary Data 4), without distinctive enrichment in

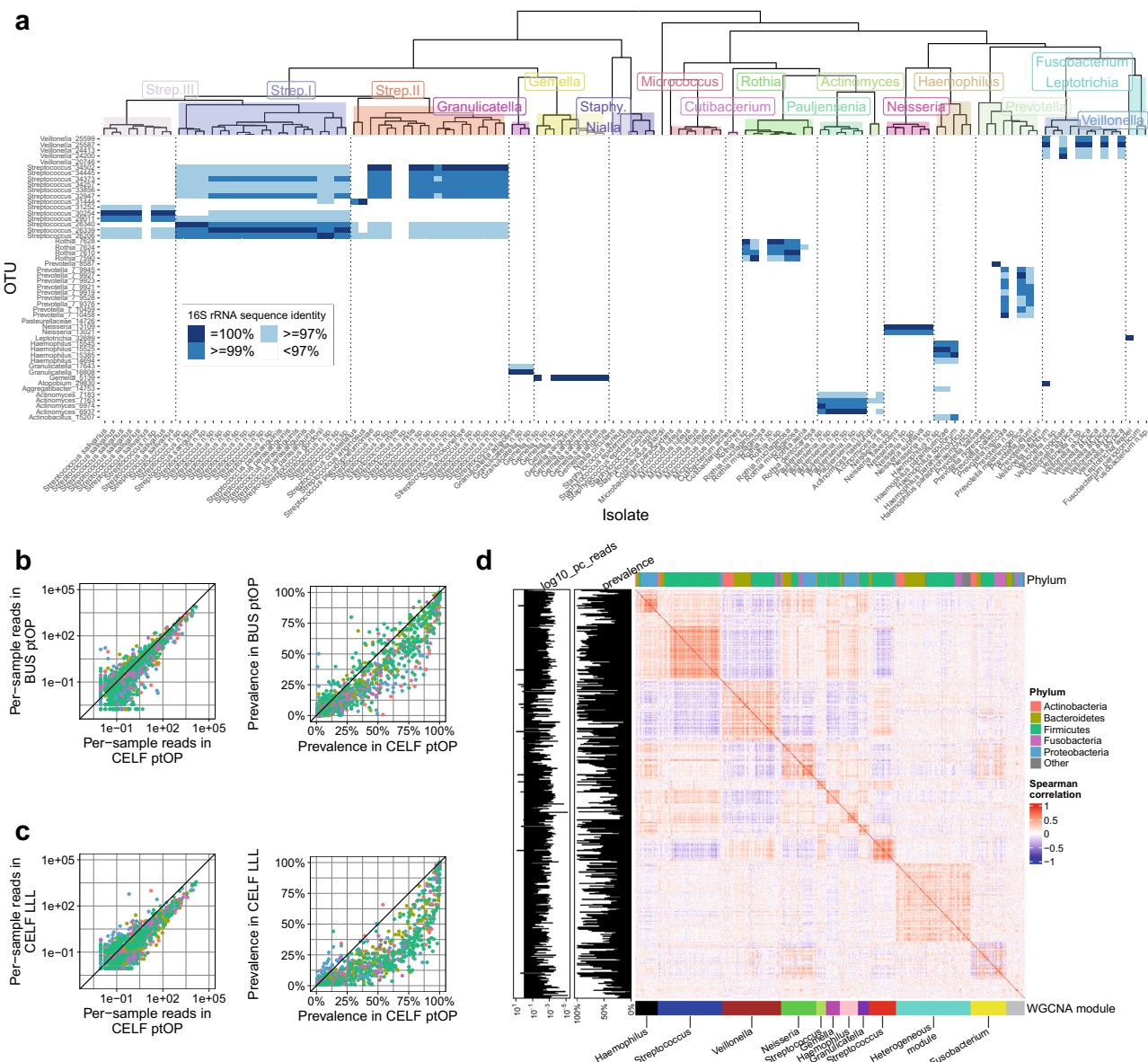

**Fig. 2 Ecology and structure of airway microbial communities. a** Mapping of the 50 most abundant OTUs onto 126 novel airway isolates. Isolates are grouped into 16 clusters according to the distance and branching order of their inferred Kegg Ontology (KO) gene content. OTU/isolate nt identity is shown as 95–97% (light blue), 97–99% (medium blue), and 100% (dark blue). The complex relationship between OTUs and isolates reflects multiple copies of the 16S rRNA gene in different taxa, but in general, captures KO phylogenetic structures. **b** Comparison of abundance (left) and prevalence right) of bacterial OTUs in populations from northern (CELF) and southern (BUS) hemispheres. The species distribution is similar between the CELF and BUS studies. **c** Comparison of abundance (left) and prevalence right) of bacterial OTUs in the posterior oropharynx (ptOP) and the left lower lobe (LLL) in CELF subjects. The relative abundance of organisms in ptOP is very similar to those in the LLL, although absolute abundance is an order of magnitude lower in the LLL. Lower abundance OTUs in the CELF dataset are more prevalent in the upper than lower airways. **d** Spearman correlations between the abundance of organisms in the CELF ptOP samples, showing a high degree of positive and negative relationships between OTUs that is the basis of WGCNA network analysis. Common phyla are colour coded at the top of the matrix, and WGCNA modules (named for the most abundant membership) are at the bottom. Network module membership may be dominated by a single phylum (e.g., the *Haemophilus* or *Streptococcus* modules) or contain mixed phyla (e.g., the *Veillonella* module).

particular genera. They included homologues of antitoxin *YefM* (57 isolates); exfoliative toxin A/B *eta*, (57 isolates); toxin *YoeB* (51isolates); antitoxins *HigA-1* (31) and *HigA* (30); antitoxin *PezA* (26); toxin *RtxA* (15); antitoxin *HipB* (14); toxin *YxiD* (13); antitoxin *CptB* (12); antitoxin *Phd* (11); and toxin *FitB* (10). These have not been previously recognised in commensal organisms and differ from the toxin spectrum of known airway pathogens[29]. They may have significant influences on the mucosa as well as other organisms.

*Nitric oxide.* Nitric oxide (NO) is a central host signalling molecule in the airways, where it mediates bronchodilation, vasodilation, and ciliary beating[30]. NO exhibits cytostatic or cytocidal activity against many pathogenic microorganisms[31] and NO elevation in exhaled breath is used as a clinical marker for lower airway inflammation. Many isolate genes encoded NO reductases (Supplementary Data 4), including *norB* (27 isolates); *norV* (11), *norQ* (5), *norC* (1) and *norR* (1). The *hmp* gene, encoding a NO dioxygenase, was present in 39 organisms. These

enzymes may mitigate the antimicrobial activities of NO or affect host bronchodilation and mucus flow.

*Iron and haem.* Iron is an essential nutrient for humans and many microbes and is a catalyst for respiration and DNA replication[32]. Host regulation of iron distribution through many mechanisms serves as an innate immune mechanism against invading pathogens (nutritional immunity)[32].

We identified 47 genes with "iron" in their KO name (Supplementary Data 2f). Those found in >75% of isolates were *afuC* (iron (III) transport system ATP-binding protein), *ABC.-FEV.P* (iron complex transport system permease protein), *ABC.FEV.S* (substrate-binding protein), and *ABC.FEV.A* (ATP-binding protein). A further 19 genes were identified as members of "haem" pathways (Supplementary Data 4).

*Haemophilus* spp. require haem for aerobic growth and possess multiple mechanisms to obtain this essential nutrient. These genes may play essential roles in *Haemophilus influenzae* virulence[33]. In our isolate collection *sitC* and *sitD* (manganese/iron transport system permease proteins) and *fieF* (a ferrous-iron efflux pump) were only found in *Haemophilus* spp., as were *ccmA, ccmB, ccmC, ccmD* (haem exporter proteins A, B, C and D) and *hutZ* (haem oxygenase). These are potential therapeutic targets.

*Sphingolipids.* The sphingolipids constitute an important class of bioactive lipids and include ceramide and sphingosine-1-phosphate (S1P). Ceramide is a hub in sphingolipid metabolism and mediates growth inhibition, apoptosis, differentiation, and senescence. S1P is a key regulator of cell motility and proliferation[34].

Sphingolipids play significant roles in host antiviral responses[35,36] and resistance to intracellular bacteria[37]. Their importance in humans is exemplified by a major childhood asthma susceptibility locus that upregulates *ORMDL3* expression[38]. ORMDL3 protein acts as a rate-limiting step in sphingolipid synthesis[39] and the *ORMDL3* locus greatly increases the risk of HRV-induced acute asthma[40].

De novo synthesis of sphingolipids is recognised in human bowel bacteria[41] and maintains intestinal homoeostasis and microbial symbiosis[42]. In the skin, commensal *S. epidermidis* sphingomyelinase makes a crucial contribution to skin barrier homoeostasis[43]. Based on KO annotations, we did not find obvious SPT homologues in our isolates but identified 12 genes with putative roles in sphingolipid metabolism (Supplementary Data 4). Of these, *SPHK* (sphingosine kinase, present in 12 isolates) which metabolises sphingosine to produce S1P; and *ASAH2* (neutral ceramidase, present in seven isolates) have potential roles in modifying host inflammation and repair. These may interact with the *ORMDL3* disease risk alleles described above.

*Immune inhibition.* Several genes present in the isolates may directly affect host immunity. These were enriched in *Prevotella* spp. (Supplementary Data 4) and included immune inhibitor A (*ina*), a neutral metalloprotease secreted to degrade antibacterial proteins; *Spa* (immunoglobulin G-binding protein A), *sbi* (immunoglobulin G-binding protein Sbi); *omp31* (outer membrane immunogenic protein); *blpL* (immunity protein cagA); and *impA* (immunomodulating metalloprotease).

A conserved commensal antigen, β-hexosaminidase (HEXA_B), has a major role in induction of anti-inflammatory intestinal T lymphocytes[44], and is present in 59 of our isolates with enrichment in *Prevotella*, *Streptococcus* and *Pauljensenia* spp.

*Autoantigens.* Systemic lupus erythematosus (SLE) and Sjögren syndrome are chronic autoimmune inflammatory disorders with multiorgan effects. Lung involvement is common during the course of the disease[45]. Our *Neisseria* isolates contain a 60 kDa SS-A/Ro ribonucleoprotein (Supplementary Data 4) that is an ortholog to the human *RO60* gene, a frequent target of the autoimmune response in patients with SLE and Sjögren's syndrome.

Other bacterial genomes contain potential Ro orthologs[46], and a bacterial origin of SLE autoimmunity has been suggested[47]. Here, the abundance of *Neisseria* spp. in human airways and their close proximity to the mucosa are of interest, as is a recent report that the lung microbiome regulates brain autoimmunity[48], and an earlier observation that T cells become licensed in the lung to enter the central nervous system[49].

It is relevant that products of cognate microbial-immune interactions in the airways have direct access to the general arterial circulation through the left side of the heart, whereas molecules and cells carried in venous blood from the gut undergo extensive filtration and metabolism in the liver before accessing more distant sites.

*CRISPR genes.* Most respiratory viruses, including SARS2-Cov-19, have RNA genomes, and RNA-targeting CRISPR vectors have the potential to prevent or treat viral infections[50]. Type III RNA-targeting system elements (such as *cas10, cas7, csm2* and *csm5*)[51] are present in our isolates (particularly *Fusobacteria* and *Prevotella* spp.), as is the Type II system element *cas9* (Supplementary Data 4).

### Isolates in the context of airway communities

*Community coverage.* We sought context for our culture collection within the ecological variation of different geographic and anatomical locations. We studied airway microbial communities in 66 asthmatics and 44 normal subjects recruited from centres in Dublin (48 subjects), Swansea (46 subjects) and London (16 subjects) (collectively known as the Celtic Fire Study (CELF)). Swabs were taken from the posterior oropharynx (ptOPs) and bronchoscopic brushings from the left lower lobe (LLL) in all subjects. When tolerated, the left upper lobe (LUL) was also brushed in 52 subjects. We compared the European CELF microbial communities to 527 ptOP samples from an adult population sample in Busselton, West Australia (BUS)[18]. Operational Taxonomic Units (OTUs) were identified by sequencing the 16 S rRNA gene amplicon and compared with the assembled genomes from our culture collection.

In the CELF ptOP samples, 17 operational taxonomic units (OTUs) covered >70% of the abundance and 41 OTUs covered >85% (Supplementary Data 5). Coverage was less in LLL and LUL samples (respectively 64% and 50% at the 70% threshold), due to the expansion of *H. influenzae* (OTU Haemophilus_14694) and *Tropheryma whipplei* (OTU Glutamicibacter_5653) in the pulmonary samples, particularly those from asthmatics (Supplementary Data 5).

Fifteen of the 17 most abundant OTUs were mapped to at least one isolate using a 99% nucleotide (nt) identity, and eleven of the next 24 OTUs were mapped to a cultured organism. Genera of moderate abundance (2.8%-0.4% of the total) yet to be cultivated include *Fusobacterium, Selenomonas, Alloprevotella, Porphyromonas, Leptotrichiaceae, Megasphaera, Lachnospiraceae, Solobacterium,* and *Capnocytophaga*.

OTUs corresponding to isolates for *Staphylococcus, Micrococcus* and *Cupriavidus* spp. had minimal representation in the community OTU analyses, although *Staphylococcus aureus* is a recognised lung pathogen. Their appearance in the isolates may represent oral or skin contamination or assertive growth in culture.

Mapping of the 50 most abundant OTU sequences onto the 126 isolates revealed complex relationships that reflect multiple copies of the 16 S rRNA gene in different taxa[52] (Fig. 2a). In general, however, OTU assignment reflected the principal KO phylogenetic structures and referencing of OTU communities to our isolate genomes may still inform on community functional capabilities.

The 16 S rRNA gene sequences poorly detected the extensive diversity of *Streptococcus* spp. in airways, as noted previously[18]. However, combinations of OTUs can be seen to form "barcodes" (Fig. 2a) that may refine *Streptococcus* spp. identification into their three main KO phylogenetic groups.

*Biogeography and community structure.* The taxa defined by OTUs and their relative abundances were similar in CELF ptOP and CELF LLL samples and to the normal population in BUS ptOP (Fig. 2b, c). Other than the most abundant organisms, the prevalence of most OTUs was lower in the LLL than in the ptOP (Fig. 2c). The mean bacterial burden was much higher in ptOP samples from CELF than in the LLL (log10 mean 7.86 ± 0.07 vs 5.06 ± 0.05), consistent with previous studies[8,16,17].

Strong correlations and anti-correlations were present between the abundances of OTUs in data from each site (exemplified for CELF ptOP samples in Fig. 2d, and previously shown for the BUS ptOP results[18]). We used WGCNA analysis to find networks (named arbitrarily with colours) within these correlated taxa. Network structures were consistent in the CELF and BUS ptOP communities (Supplementary Figs. 3 and 4), but less distinct in the lower airway samples where taxa were less diverse and of lower abundance (Supplementary Fig. 5).

Networks often contained closely related species but also extended beyond phylogenetically related organisms (Fig. 2d and Supplementary Fig. 6). For example, in the CELF ptOP networks (Fig. 2d and Supplementary Fig. 6) there are phylogenetically homogeneous modules of *Streptococci* (blue, red and green-yellow), *Gemella* (magenta), *Haemophilus* (black and pink) and *Granulicatella* (purple).

Of interest is the brown module in the CELF ptOP samples, which contains multiple *Prevotella* and *Veillonella* spp. of high abundance. The presence of biofilm elements in *Prevotella* spp. described above supports a hypothesis that these organisms may adhere to form a basic "commensal carpet" of the airways[18].

Both the CELF ptOP and BUS ptOP networks recovered the phylogenetic relationships found in the KO analysis amongst *Streptococcus* isolates. The three clusters of *Streptococcus* isolates (Strep. I-III) map to distinct sets of OTUs using sequence similarity (Fig. 2a), and this similarity is also uncovered in the WGCNA network modules in both ptOP networks (Supplementary Fig. 7).

*Dysbiosis.* Subtle alterations in bacterial community composition ("dysbiosis"[53]) are recognised in many diseases with microbial components. Community instability and inflammation in the presence of mild viral infections[5] should be added to the recognised features of loss of diversity and pathobiont expansion in asthma and COPD. We, therefore, sought insights into airway dysbiosis in our subjects from genomic sequencing of the commensal organisms.

We explored underlying components of airway communities by using Dirichlet-Multinomial Mixtures (DMM)[54] on all samples from the BUS and CELF subjects, finding that samples formed predominantly into two clusters (Airway Community Type 1 and 2: ACT1 and ACT2) (Fig. 3a). The main drivers for the two pulmotype clusters were identified as *Streptococcus*, *Veillonella*, *Prevotella* and *Haemophilus* spp. in descending order of relative abundance across all samples. ACT1 was dominated by *Streptococcus*, *Veillonella* and *Prevotella* in 410 samples; whilst ACT2 was dominated by *Streptococcus*, *Veillonella* and *Haemophilus* in 478 samples (Fig. 3a). Principal coordinates analysis based on Bray-Curtis-distance (β-diversity) of the airway microbiota confirmed significant overall compositional differences between the two community type clusters (PERMANOVA $P$-value > 0.001) (Fig. 3b).

Congruence analysis of CELF samples (Fig. 3c) confirmed consistency in assignment for samples coming from the same donor ($\chi^2 < 0.005$) or the same sampling site ($\chi^2 < 0.005$).

We performed univariate analysis to investigate the association between CELF subject metadata and potential indicators of dysbiosis, specifically, evenness and richness (Fig. 3d), and bacterial abundance at the phylum level (Fig. 3e). Features describing clinical phenotypes and sample origin were often strongly collinear. We, therefore, assessed found associations in turn for retained significance with each potential confounder, using a nested rank-transformed mixed model test[55] and considering repeated sampling of patients as a random effect.

We saw pervasive effects both on alpha diversity and phylum level of the tested predictors (Fig. 3d, e). Importantly, the Shannon index and richness were significantly decreased with asthma status and severity (MWU false-discovery rate (FDR) < 0.1) (Fig. 3d).

We found an increase (although not significant) of the *Proteobacteria* Phylum associated with asthma status (Fig. 3e), in line with the taxonomic profile of patients with asthma vs. healthy controls (Fig. 3g). This is consistent with many reports of *Proteobacteria* excess in asthmatic airways[8,9,56]. Type 2 communities were enriched in subjects with positive asthma status in all sample sites and in CELF subjects overall (Fig. 3f).

We examined the impact of the study, asthma status, and sampling site on the distribution of community types in the CELF thoracic samples, using logistic regression models with sex and age as control variables. The results indicated significant differences in ACT proportions across different sampling sites: LUL vs. OTS: odds ratio 95% confidence interval 0.135–0.444 ($p$-val: 3.1e-07); LLL vs. OTS: 0.049–0.249 ($P$-val: 5.0e-10). Statistical significance was more marked for the left upper lobe (FDR $q$-value < 0.001) than the left lower lobe ($q < 0.10$).

We extrapolated metabolic activities from binning 16S rRNA gene abundance onto the isolate KOs using PICRUSt[57], revealing metabolite profiles that distinguished measures of diversity and location within upper or lower airways (Fig. 3h), as well as distinctive features of asthma and dysbiosis.

*Mucosal factors.* In order to relate our mapped microbiome to its ecosystem, we sought host components of the microbial-mucosal interface by serial measurements of global gene expression and supernatant metabolomics during full human airway epithelial cell (HAEC) differentiation in an air-liquid interface (ALI) model. We hypothesised that the transition from monolayer to ciliated epithelium over 28 days would be accompanied by the progressive expression of genes and secretion of metabolites for managing the microbiota.

HAEC from a single donor were grown in triplicate and harvested on days 0, 2, 3, 7, 14, 21 and 28. Trans-epithelial resistance (TEER) rose from 7.4 ± 0.3 on day 0 to 1551 ± 113 on day 28, and MUC5AC mRNA production rose 30-fold over the same period (Supplementary Fig. 8), indicating full epithelial development.

We found 2553 significantly changing transcripts organised into eight core temporal clusters of gene expression (Limma, 3.22.7) (Fig. 4a and Supplementary Data 6). Late peaks of expression were found in four clusters, three of which (CL2, CL4 and CL5) contained many genes likely to interact with the

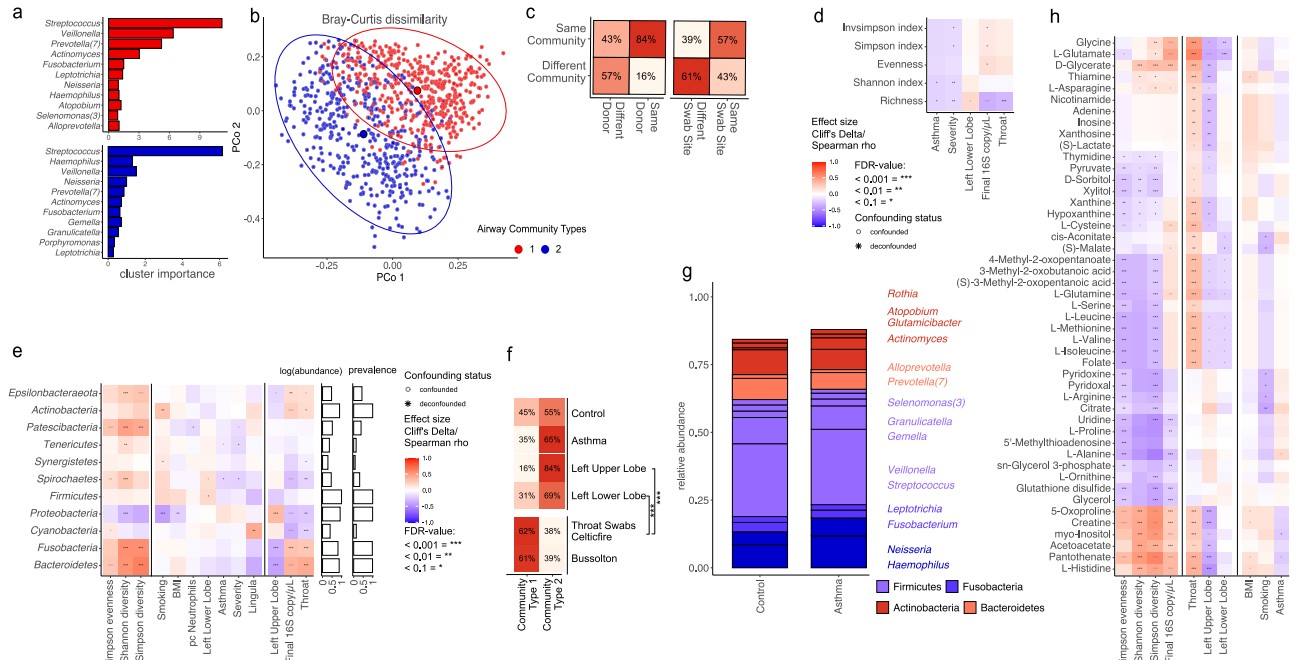

**Fig. 3 Microbial features of airway dysbiosis. a** Main drivers of Dirichlet-multinomial model-based airway communities. **b** Beta diversity based on Bray-Curtis dissimilarity principal coordinate analysis showing separation of the two communities. **c** Consistency of airway community assignment between samples of the same and different donors (left) and sampling sites (right). **d** Alpha diversity measures and correlations. **e** Univariate associations of CELF 16S samples binned on phylum level to metadata. **f** Proportion of community assignments between ptOP samples of different study origins, sampling sites and disease groups. **g** relative abundance of most abundant genera based on CELF samples 16S rRNA. **h** Univariate metabolite associations based on binning of CELF 16S rRNA sequences onto isolate annotation.

microbiome (Supplementary Data 6). Transcripts in the other upgoing cluster (CL3) were elevated early and late in differentiation and were enriched for genes mediating cell mobility and localisation. Genes of particular interest in the other upgoing clusters are as follows.

*Mucins and ciliary development.* Mucosal mucins are central to mucosal function and integrity, providing a source of nutrients and sites for tethering of commensals[58], whilst restricting the density of organisms through upward flow by beating cilia[59]. Interactions of mucins with microbiota play an important role in normal function[58], and direct cross-talk between microbes and mucin production is likely[59].

In our ALI model, progressive up-regulation of the major secreted respiratory mucins *MUC5AC* and *MUC5B* in CL2 was accompanied by the membrane-associated *MUC20* (Supplementary Data 6). In contrast, CL5 contained three membrane-associated mucins (*MUC13*, *MUC15*, *MUC16*). These mucins do not form gels and are anchored to the apical cell surface, where they present a glycoarray for selective interactions with the microbial environment[58].

Within CL5 we also found 17 gene families and 175 genes with putative roles in ciliary function, ciliogenesis, or spermatogenesis (Supplementary Data 6). Mutations in many of these genes are known to cause primary ciliary dyskinesia (PCD)[60], which results in recurrent pulmonary infections. Other genes in this list are candidates for mutation in cases of PCD without known cause.

*Immune-related genes.* The most significant effects (top hits) in CL2 included *ENPP4* (which promotes haemostasis); *ALOX15* (which generates bioactive lipid mediators including eicosanoids); *GLIPR2* (which enhances type-I IFNs); *MPPED2* (a metallophosphoesterase active in infection); *INSR* (insulin receptor); and *MIR223* (an inhibitor of neutrophil extracellular trap (NET) formation in infection) (Supplementary Data 6).

Immune-related genes significantly expressed in CL5 included complement factor 6 (*C6*) which forms part of the membrane attack complex. C6 deficiency is associated with *Neisseria* spp. infections. *CD38* was also highly expressed, and its product is an activator of B-cells and T-cells.

*Detoxification and transportation.* Top hits in CL4 include *ADH1C*, an alcohol dehydrogenase; *GSTA2* with a known role in the detoxification of electrophilic carcinogens, environmental toxins and products of oxidative stress by conjugation with glutathione; *ACE2*, the SARS2-Cov-19 binding site which cleaves angiotensins; and *PIK3R3* which phosphorylates phosphatidylinositol to affect growth signalling pathways (Supplementary Data 6).

CL4 contains five members of the cytochrome P450 families with potential roles in the detoxification of microbial products, including *CYP2F1* (which modifies tryptophan toxins and xenobiotics); *CYP4X1* (unknown substrates); *CYP4Z1* (benzyl esters); *CYP4F3* (Leukotriene B4); and *CYP2C18* (sulfaphenazole). Also in CL4 were transporters *SLC10A5* (substrate bile acids); *SLC27A2* (fatty acids); *SLC1A1* (glutamate); *SLC4A11* (borate); *SLC25A4* (ADP/ATP in mitochondria); *SLC45A4* (sucrose); *SLC25A28* (iron); and *SLC39A11* (zinc).

Enrichment of genes for detoxification and transport was also present within CL2, which included *CYP4B1* (substrate fatty acids and alcohols); *CYP4V2* (fatty acids); *CYP2A13* (nitrosamines); *CYP2B6* (xenobiotics); *CYP26A1* (retinoids); and *CYP4F12* (arachidonic acids). Transporters included *SLC40A1* (iron); *SLC13A2* (citrate); *SLC15A2* (small peptides); *SLC12A7* (KCl co-transporter); and *SLC35A5* (nucleoside sugars).

*Neuronal development.* The bronchial mucosa is innervated with vagal sensory unmyelinated fibres that detect airway luminal substances and mediate smooth muscle tone, mucus secretion, and cough[61]. Airway sensory nerves are directly involved in

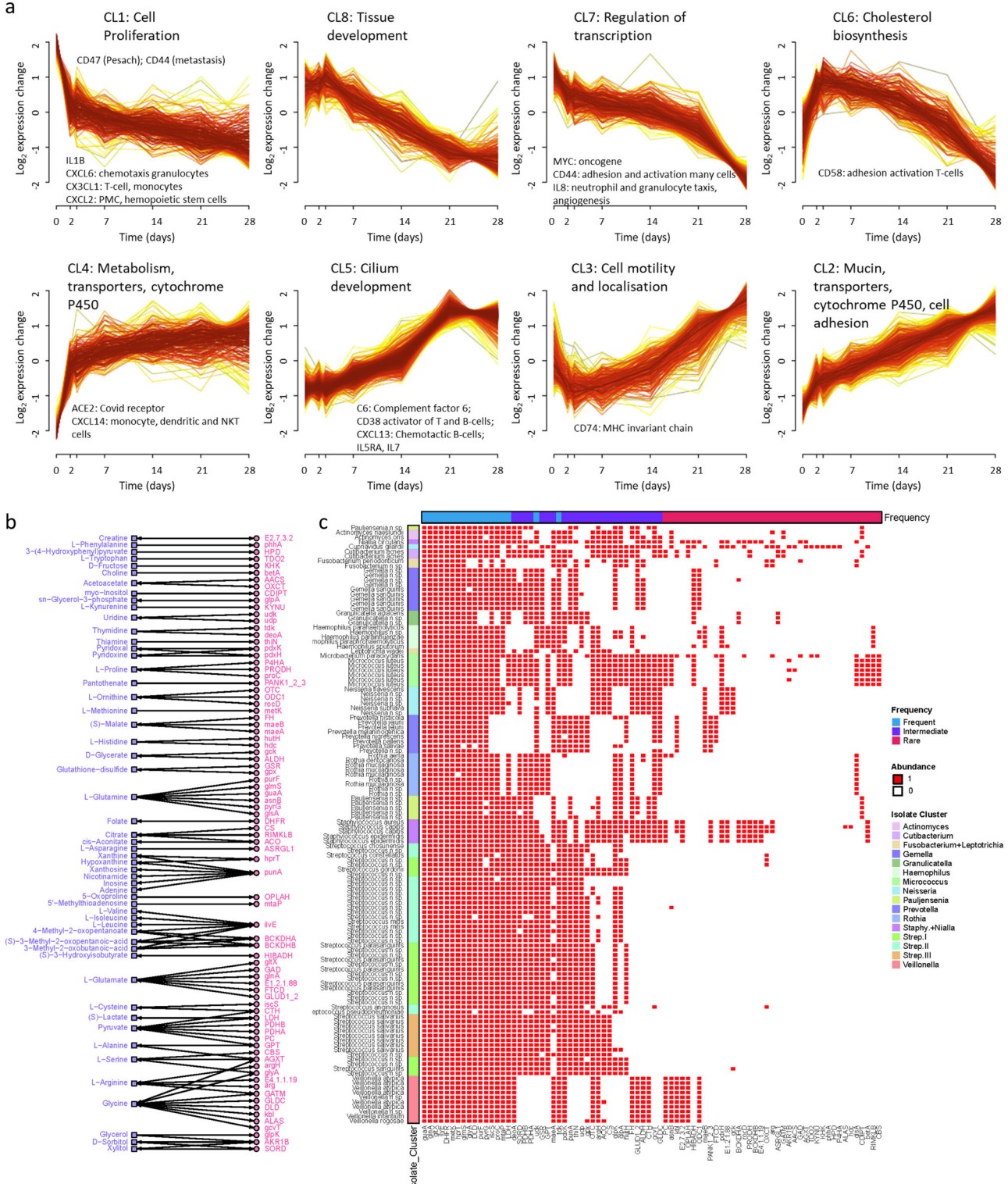

**Fig. 4 Gene and metabolite abundance during airway epithelial development. a** Global gene expression was measured 7 times over 28 days in an air-liquid model of epithelial differentiation (monolayer to ciliated epithelium). A total of 2,553 transcripts, summarised by 8 core temporal profiles, showed significant variation in abundance during mucociliary development. Hallmark functional roles are shown for each cluster. Clusters CL2, CL3, CL4 and CL5 show late peaks of expression and contain genes that can interact with the microbiome. Upregulated chemokines and immune-function genes are also noted within the clusters. **b** Metabolites (square) measured in the supernatant of the fully differentiated airway cells were linked to genes (circle) identified in bacterial isolates. Arrows indicate if the reactions were reversible or irreversible, with metabolites as substrates and products. These networks were built based on KEGG pathways. **c** Binary heatmap displaying the presence (1) or absence (0) of genes (columns) identified in the genomic sequences of bacterial isolates (rows). Bacterial isolates are organised into Kegg Ontology phylogeny clusters (see Fig. 2). Gene annotations (top) indicate the frequency of the gene: 'frequent' for genes in >75% of isolates, 'intermediate' for genes in 25–75% of isolates and 'rare' for those in <25% of isolates.

immune or inflammatory responses, themselves releasing proinflammatory molecules ("neurogenic inflammation")[62,63]. Neuroinflammation can change receptors, ion channels, neurochemistry, and fibre density[64]. It contributes to the disabling syndrome of cough hypersensitivity and chronic cough[65].

A basis for innervation can be seen in top hits from CL2, which included *ENPP5* and *HECW2*, which have putative roles in the development of airway sensory nerves (Supplementary Data 6). Interestingly, CL2 and CL4 together contained ten members of the protocadherin beta gene family (*PCDHB2*, *PCDHB3*, *PCDHB4*, *PCDHB5*, *PCDHB10*, *PCDHB12* and *PCDHB18P* in CL2; *PCDHB13*, *PCDHB14*, and *PCDHB15* in CL4). Interactions between protocadherin beta extracellular domains specify self-avoidance in specific cell-to-cell neural connections[66], and their abundant presence here may regulate singular neural-mucosal cell coherence.

*Intersection of mucosal and microbial metabolomic pathways.* Metabolites are central to biological signalling, and so we used the same time-series model of AEC differentiation to measure levels of metabolites released into the culture media of the cells (Supplementary Data 7).

We then mapped the ALI culture metabolites to enzymes in matching bacterial pathways identified within the KO of isolate genomes (Fig. 4b), based on direct reactions, as substrates or products. Notable interactions include amino acids, nucleotides and compounds involved in energy metabolism. The metabolite-related KOs exhibited distinctive patterns within the isolate phylogeny (Fig. 4c).

Enrichment of these KOs onto global human and bacterial KO pathways with iPath[67] is shown in Supplementary Figs. 9 and 10. These suggest folate biosynthesis is ubiquitous amongst airway organisms, valine, leucine and isoleucine metabolism to be of intermediate importance and alanine, aspartate and glutamate metabolism to be rare functions amongst the isolates.

## Discussion

Our results describe the systematic culture, isolation and sequencing of the respiratory commensal bacteria. Although the principal airway phyla are well known through OTU studies of whole communities, previous attempts at culture have been limited to patients with Cystic Fibrosis (CF)[68–70], a disease in which CFTR mutations induce major changes in the airway mucosal fluid and host environment. Anaerobic species cultured from these studies include the genera *Actinomyces*, *Atopobium*, *Micrococcus*, *Neisseria*, *Prevotella*, *Rothia*, *Streptococcus*, and *Veillonella*[69], and may be similar to our isolates. Nevertheless, systematic commensal sequencing has not previously been carried out, and 40% of our isolates are novel species. Their gene content indicates a wide range of previously undocumented capacities to interact with other organisms and the airway mucosa.

*Streptococcus* species showed the greatest novelty, with 60% of isolates not previously found in public databases. These are in phylogenetic clusters distinct from known respiratory commensals such as *S. salivarius* and *S. parasanguinis*. Their abundance in the oropharynx and lower airways suggests important functions that are yet to be explored.

Our findings mean that it is now possible to investigate systematically the effects of individual bacteria and their combinations on airway inflammation and infection. Therapies derived from healthy microbial communities are established for inflammatory and metabolic bowel diseases, through faecal transplantation, bacteriotherapy with specific organisms[71], and bacterial metabolites[72]. Inhibition of inflammation in airway epithelial cell models has recently been shown for *Rothia*, *Prevotella* and *Streptococcus* spp. grown from children with CF[69,70].

cRich microbial environments are well known to protect against asthma in schoolchildren[73] and adults[74], although the responsible organisms have not been identified in airway communities. We have previously found reduced numbers of *Selenomonas*, *Megasphaera* and *Capnocytophaga* spp., in asthmatic ptOP samples[18]. Despite their moderate abundance (0.4-2.8% of the total) we have not managed to culture them. Future isolation is desirable to test if they are indicator species or direct contributors to respiratory health.

Lower respiratory tract infections are the fourth most common cause of death globally. In the UK alone 16 million UK patients with respiratory infections are treated with antibiotics annually[75], a major driving force in antimicrobial resistance (AMR)[75,76]. Genomic sequences from our isolates and the negative abundance correlations in airway communities indicate the presence of "natural" antimicrobial factors that can now be systematically identified with therapeutic intent.

The large number of novel *Streptococcus* spp. in our isolates and the poor OTU discrimination of *Streptococcus* spp. confirm that 16S rRNA gene sequences fail to identify much of the diversity in this genus[18,77], which is expanded in severe asthma[78] and in heavy smokers[18]. OTU analyses have also failed to identify abundant novel species identified as *Pauljensenia* by our genomic sequences, assigning them instead to *Actinomyces* spp. Our isolates will support the detailed investigation of these poorly understood genera.

Metagenomic and metatranscriptomic sequencing has been very informative in understanding bowel microbial activities in health and disease. In contrast, non-purulent airway secretions typically contain <5% microbial DNA[79] and are difficult to access. Purulent secretions, such as sputum, are often heavily contaminated with upper airway and oral flora[20]. Consequently, metagenomic sequencing of respiratory samples has so far identified only the most abundant pathogens and commensals, with limited functional resolution[20,79,80]. By extending available airway genome and gene catalogue data as we have here, sequenced reads too sparse to reliably assemble per sample can be mapped to our gene and genome assemblies. This will provide a scaffold for metagenome analyses as well as for the selection of marker genes and primers adapted for targeted amplicon sequencing of specific airway microbiota. As shown above, the gene content of airway communities can also be inferred by mapping genome sequences to OTU results. Thus, through the present collection, taxonomic and functional characterisation of airway communities is facilitated.

We have studied HAEC from a single donor, and it is to be expected that multiple genetic and epigenetic factors will influence different components of the pathways we have identified. Such factors may in the future be systematically investigated by knockdown and knock-in in model systems and by the culture of HAEC from subjects with and without airway diseases[81]. It is already clear that the co-culture of pathogens and commensals in such models will reveal many further pathways underpinning host-microbial interactions[69,70].

Microbial community dysbiosis with diversity loss and overgrowth of pathobionts is recognised in asthma, COPD and other pulmonary disorders[9,19]. HRV infections are the major precipitant of acute exacerbations of asthma[82,83] and of COPD[84,85] yet have trivial effects in most individuals. Here we have found networks of interacting bacteria that are attenuated in the lower airways, possibly presaging loss of stability[86]. The hypothesis can now be tested that microbial community instability predisposes to dysregulation of inflammatory processes during acute exacerbations of lung disease.

## Methods

**Microbial culture**. After sampling, bronchial brushes for extended culture were immediately placed in 15 ml centrifuge tubes with 2 ml sterile saline solution (0.9% w/v) and immediately transported to the laboratory for processing. Samples were mixed on a vortex mixer twice for 5 s. On duplicate plates, 100 μl of the saline was plated on Columbian blood agar (5% horse blood), chocolate agar, or minimal agar with 0.5% (w/v) mucin. One set of plates was incubated at 37 °C in a standard atmosphere while the other set was incubated at 37 °C in an anaerobic workstation (Don Whitley DG250). Colonies were selected from 24 h to 168 h by appearance, streaked out on their corresponding media and incubated for a minimum of 48 h. Plates were then colony-selected again and Gram-stained. Aerobic isolates were tested for oxidase and catalase activity. DNA was extracted from brain heart infusion broth for aerobes and sodium thioglycollate media for the anaerobes. Isolates that failed to grow in liquid medium were grown on solid medium and an inoculation loop was used to scrape growth off the surface of the agar prior to DNA extraction.

**Whole-genome sequencing bacterial isolates**. Whole-genome sequencing was carried out at the Wellcome Sanger Institute, using the HiSeq X platform and generating paired-end read lengths of 151 bp. Genomes were de novo assembled using Bactopia[87] (v 1.4.11). Taxonomic classification and quality control were performed using MiGA (http://microbial-genomes.org/) with the TypeMat database. Isolates appearing to contain multiple genomes were discarded.

For all assemblies, the average nucleotide identity was computed using fastANI[88] (v 1.3) with a fragment length of 500 bp and clustered on 99.5% average nucleotide identity. For every cluster, sequencing data of every entity (isolate) were pooled and processed using Bactopia (v 1.4.11) with default settings. Taxonomic annotation and novelty scores were computed using MiGA with the TypeMat database as well as the NCBI Prokaryote genome database for comparison. Functional annotation was performed using prokka (v 1.14.6) as implemented in Bactopia; and eggnog-mapper[89] (v emapper-1.0.3-40-g41a8498) using diamond (v 0.9.24) for the alignments, reducing the search space to the domain of bacteria. Antimicrobial resistances were annotated using amrfinder (v 3.8.4) and ARIBA (v 2.14.5) using the CARD database (v 3.0.8). Virulence factors were computed using the VFdb core dataset (v) and binned into higher functional entities using a custom perl script.

Phylogenetic analysis of the isolates was performed using the Bacsort pipeline (https://github.com/rrwick/Bacsort). First, fastANI distances were computed with a fragment length of 1,000 bp and a maximum distance of 0.2. A phylogenetic tree was constructed using as implemented in the R-package ape[90] (v 5.6-2). The tree was visualised using the Interactive Tree of Life (iTol)[91]. Small ribosomal subunits were extracted from assembled genomes using Metaxa2 and aligned with CELF OTUs using BLAST with 100% percentage nucleotide identity, e-value = 1e-10, and length ≥ 206 bp.

**Kegg Ontology and isolate phylogeny**. From the eggnog-mapper output, we derived 5531 Kegg Ontology (KO) annotations for the 126 isolates which we encoded in a binary matrix indicating presence/absence. We removed 254 zero-variance KOs (that were either present in all or no isolates) and performed hierarchical clustering of the isolates with the 5023 remaining KOs using the Manhattan distance metric and complete linkage. The distance matrix was calculated after removing 2313 KOs that had identical presence/absence to at least one other isolate. The distance matrix was calculated after removing 2313 KOs that had identical

presence/absence to at least one other isolate. The Dynamic Tree Cut algorithm[27] identified 15 clusters of isolates that recovered known phylogenetic relationships (Fig. 2a). These 15 clusters were then mapped to the OTUs using the 16S rRNA gene sequence similarity (Fig. 2a). Based on OTU similarities, one *Streptococcus* cluster was split into two additional clusters, resulting in a final set of 16.

We then identified characteristic KOs that were over- or under-represented in each cluster relative to all other clusters. We scored cluster i and KO j using a 2 × 2 contingency table, where a: number of isolates in cluster i containing KO j; b: number of isolates in cluster i without KO j; c: number of isolates not in cluster i containing KO j and d: number of isolates not in cluster j without KO j; from which we calculated odds ratios (ORs) using ad/bc. 0.5 was added to cells with zero counts (the Haldane-Anscombe correction). Log10(OR) was used as a summary statistic to rank the KOs by importance for a given cluster. The 2313 duplicate KOs were assigned the same score as their duplicated counterpart used to construct the distance matrix.

**Human study populations**. Samples included in this study were collected from two study populations, The microbial pathology of asthma study (Celtic Fire, CELF) and the Busselton Health Study, a long-running epidemiological survey in South-Western Australia (BUS).

The CELF study was a multicentre, cross-sectional study of asthmatic adults and healthy controls. Participants were recruited from 3 UK centres, Connolly Hospital, Dublin; The Royal Brompton Hospital, London; and Swansea University Medical School, Swansea. Ethical approval for the study was granted by the London-Stanmore Research Ethics Committee (reference 14/LO/2063). All subjects provided written informed consent. Subject groups were: healthy subjects (non-smokers and current smokers; asthmatic patients taking short-acting beta-agonists only (BTS Step 1); asthmatics on moderate dose of inhaled corticosteroid (ICS) (up to 800 μg/day of beclomethasone propionate (BDP equivalent) ± long-acting β-agonist LABA (BTS Step 2/3); asthmatics on high dose ICS (ICS dose >=1600 μg/day) + LABA ± other controllers (theophyllines, LTRA, LAMA) (BTS Step 4); and asthmatics on high dose ICS (ICS dose >=1600 μg/day) + LABA ± other controllers + oral prednisolone ± anti-IgE treatment (BTS Step 5). Severe asthma was defined as BTS step 4 or 5. Exclusion criteria were: Asthmatic subjects must be non-smokers or ex-smokers with <5 pack-years smoking; BMI > 35; diagnosis of rheumatoid arthritis, allergic bronchopulmonary aspergillosis, or Churg-Strauss syndrome; drug therapy with beta-blockers, ACE inhibitors, anti-asthma immune modulators other than steroids; antibiotics within 4 weeks of study; acute exacerbations of asthma within past 4 weeks; history of an upper or lower respiratory infection (including common cold) within 4 weeks of baseline assessments; confounding occupations (such as baking); and significant vocal cord disorder.

Participants were invited to initial assessments prior to bronchoscopy. A posterior oro-pharyngeal (ptOP) swab was taken from each participant immediately before the bronchoscopy commenced. During bronchoscopy, two bronchial brushings were taken from the left lower lobe (LLL) of each subject. If tolerated, two further brushes were taken from the left upper lobe (LUL). An additional bronchial brush from the left lower lobe of five study participants from The Royal Brompton Hospital were processed for extended bacterial culture (described above). Scope control washes were taken at each bronchoscopy.

All non-biopsy samples were stored at −80 °C within 1 h of collection. Those harvested at The Royal Brompton Hospital were

transported and stored directly at the Asmarley Centre for Genomic Medicine (ACGM) at the same site. Samples at other sites were stored locally at −80 °C for a maximum of 6 months prior to transport to the ACGM on dry ice.

Investigation of the BUS subjects was as previously described[18]. ptOP swabs were collected with the same protocols as CELF from 527 individuals. After local storage at −80 °C, ptOP swabs were transported on dry ice to the ACGM for further processing.

**DNA extraction and quantification**. Microbial DNA extraction from Celtic Fire samples was carried out using a hexadecyl-trimethylammonium bromide (CTAB) and bead-beating double extraction using phase lock tubes. Bacterial isolates were extracted using a single extraction method. Full details of extraction protocols for each sample type are outlined in https://doi.org/10.17504/protocols.io.bf28jqhw (Protocols.io). Busselton ptOP swabs were extracted using the MPBio DNA extraction kit for Soil, as previously described[18]. DNA was stored at −20 °C until processing. Microbial DNA quantification was carried out using a SYBR green 16S rRNA gene qPCR[92].

Within a Class 2 biological safety cabinet, each bronchial brushing was transferred directly into an LME tube. To control for contamination an empty LME tube (i.e., an extraction control) was added to each batch. The extraction control underwent the entire extraction process along with the samples. Eighteen two randomly selected Scope Control Washes (SCWs) also underwent DNA extraction.

**Microbial 16S rRNA analyses**. 16 S rRNA gene sequencing was performed on the Illumina MiSeq platform using dual barcode fusion primers and the V2 500 cycle sequencing kit. Sequencing was performed for the V4 region of the 16 S rRNA gene as previously described[18,92]. Sampling and extraction controls, PCR negatives and mock communities were included in all sequencing runs.

All samples and controls from both the Celtic Fire and BUS datasets were included in this analysis and were processed through the QIIME 2.0 analysis pipeline.

Sequences were quality trimmed to 200 bp using trim-galore (Version 0.6.4) and joined with a maximum of 10% mismatch and a minimum of 150 base pair overlap using joined_paired_ends.py (Version 1.9.1). Data was quality-checked using FASTX Toolkit (Version 0.0.14) prior to de-multiplexing.

Reads were dereplicated and open reference OTU clustering was performed in QIIME 2.0. Chimeric sequences were identified and removed, leaving borderline calls in the analysis. Phylogeny was aligned using mafft followed by consensus taxonomic classification. The Biom file, tre file, and taxa identifications were exported for further analysis.

Processed data was transferred to R (Version 3.6.3) and uploaded into Phyloseq (Version 1.3). Reads unassigned or assigned to Archaea at the kingdom level were removed before further analysis along with reads identified as Chloroplast or Mitochondria. All OTUs with less than 20 reads (reads present in less than <2% of the samples ($n = 1174$)) were removed from further analysis.

Contaminant OTUs were identified using Spearman's correlation between bacterial biomass and with number of reads per sample. OTUs were considered to be contaminants with a Benjamini–Hochberg corrected P-value of <0.05 and a correlation value of >0.2.

Due to the nature of the differences in the extraction and sequencing protocols between BUS and CELF studies, contaminants were investigated in the whole dataset and in CELF and BUS separately. OTUs identified using the individual datasets were removed from further analysis. The "Prevalence" method in Decontam (Version 1.6) with a threshold of 0.1 and controlling

for study, identified a further 55 OTUs contaminant OTUs associated with negative controls. All OTUs identified were checked and found to be consistent with contamination[93].

**Community analyses of 16S rRNA sequences**. OTU counts were rarefied to the size of the smallest retained sample (discarding samples with too few reads) to obtain the relative abundances of the microbiota in each sample accounting for read depths.

Univariate analysis was done using metadeconfoundR (https://github.com/TillBirkner/metadeconfoundR), relative abundances were tested for univariate associations with clinical variables, requiring Benjamini–Hochberg adjusted FDR < 0.1 and the absence of any clear confounders. Only major taxa and OTUs detected after rarefaction in at least 10% of samples were used.

Within metadeconfoundR, non-parametric tests were used for all association tests as the data was not normally distributed[55]. For discrete predictors, the Mann–Whitney test (two categorical variables) or the Kruskal–Wallis analysis of variance (more than two categorical variables) were used. For pairs of continuous variables, a non-parametric Spearman correlation test was used. Benjamini–Hochberg false-discovery rate control (FDR) was applied to control for multiple testing controlling the family-wise error rate at 10%.

Hierarchical clustering on the relative abundance profiles was used to establish grouping patterns of the different study samples, including an updated adaptation of the approach used to define "enterotypes" in the human gut, this so-called pulmotyping was performed using the Dirichlet Multinominal package, fitting a Dirichlet-multinomial model on the count matrix of genus relative abundance to classify genus abundance based on probability. Each count x in the matrix corresponds to a feature (of n features in total) in the composition observed in the replicate sample. Replicates are grouped into k groups. This parameterisation of the Dirichlet distribution for k parameters corresponds to the expected proportions of each of the features (e.g., a particular taxon) in group k, and is an intensity that is shared among all features. The hyperprior for the k parameters at the 'topmost', or most inclusive, level of the model hierarchy is another Dirichlet distribution with equal prior probability for each feature within the composition. These distributions together form a hierarchical model for relative abundances among samples used to cluster all samples into different pulmotypes. The chi-square test implemented in base R was used to test for significant differences in the resulting pulmotype distribution between samples grouped by disease status.

Redundancy-reduced isolate abundance/sample (from 16 S) and annotation isolate to KEGG KOs were used to generate a sample to KO projection. The projection was mapped to KOs involved in generating the metabolites highlighted by the ALI experiments[57], by multiplying taxon abundances with the KO presence/absence matrix to yield functional potentials and a proxy for expected metabolite turnover. MetadeconfoundR analysis of this matrix was then carried out together with clinical metadata accompanying the OTU abundance analysis.

**Airway epithelial cell culture**. Primary normal human bronchial epithelial (NHBE) cells (Promocell, Germany) derived from a 26-year old adult were grown on collagen-coated flasks using the Airway Epithelial Cell Growth Medium Kit (Promocell, Germany) supplemented with bovine pituitary extract (0.004 ml/ml), epidermal growth factor (10 ng/ml), insulin (recombinant human) (5 μg/ml), hydrocortisone (0.5 μg/ml), epinephrine (0.5 μg/ml), triiodo-L-thyronine (6.7 ng/ml), transferrin, holo (human) (10 μg/ml) & retinoic acid (0.1 ng/ml) (Promocell, Germany) and Primocin (Invivogen, France).

At passage 3, NHBE cells were seeded onto 12 mm Transwell inserts with 0.4 µm pore polyester membranes at a density of $2.5 \times 10^5$ cells/insert. Cells were maintained in ALI medium, a 50:50 mixture of ALI x2 media (Airway Epithelial Cell Basal Medium with 2 supplement packs added (without triiodo-L-thyronine and retinoic acid supplements) and 1 ml BSA (3 µg/ml)) and DMEM supplemented with retinoic acid (15 ng/ml) (Sigma Aldrich, Gillingham, UK). Cells were fed apically and basolaterally until 100% confluent, after which they were fed exclusively basolaterally with apical media removed. This was defined as 'Day 0', the start of the ALI culture. Media was changed three times a week for 28 days, at which stage full differentiation had occurred. At seven points during culture we performed trans-epithelial electrical resistance (TEER) measurements, took apical washings for ELISA measuring MUC5AC, harvested triplicate wells for gene expression microarray analysis and qPCR for MUC5AC mRNA as well as harvested quadruplicate wells and culture supernatants for metabolomics analysis. NHBE cell pellets and 200 µl basolateral supernatants were snap-frozen in liquid nitrogen and stored at −80 °C for metabolomic analysis.

All cell culture experiments were regularly tested for mycoplasma contamination using PCR Mycoplasma Test Kit I/ C (Promokine, Germany).

**Metabolomics analysis**. Metabolic profiling performed by Metabolon Inc (NC, USA) followed their standard protocols and used LC-MS and GC-MS methods. All samples were given unique identifiers and bar-coded for tracking throughout the analysis pipeline. The Metabolon LIMS system was used to extract raw data, identify peaks and process QCs. Metabolites were identified by comparing retention times, $m/z$ and chromatographic data to library entries of purified standards and recurrent unknown entities. All library matches were confirmed with interpretation software and the assigned compounds were curated. Missing values, below the limit of detection, were imputed with the lowest detected value for the corresponding variables for subsequent analysis.

Analyses were performed using R (version 4.1.1). The MetaboSignal package[94] was utilised to link media metabolites to KOs via their shortest paths, according to KEGG pathways. These pathways were filtered to display only direct reversible and irreversible reactions. Metabolites and KOs were mapped to human and microbial metabolic pathways using iPath 3.0 (https://pathways.embl.de/)[67].

**Transcriptomics of NHBE**. Approximately 200 ng total RNA (with one exception in which 100 ng total RNA was used) was prepared for whole transcriptome microarray analysis using the Ambion WT Expression kit. Purified cRNA yield was assessed using an Agilent 2100 Bioanalyzer and then taken forward for reverse transcription to yield sense-strand cDNA. A total of 5.5 µg of sense-strand cDNA was fragmented and labelled using the Affymetrix GeneChip WT Terminal Labelling Kit prior to hybridisation to the GeneChip ST2.1 Array. Microarray libraries were hybridised, washed, stained and imaged using the Affymetrix Genetitan.

Analyses were carried out in R (version 3.1.0). Raw data was imported into R and quality control was carried out using arrayQualityMetrics (version 3.20.0), detecting outlier arrays that are likely to skew data upon normalisation. Any outlier arrays were excluded and the corresponding samples were re-processed and run on arrays until all samples had successfully passed quality control. QC-passed arrays were normalised by Robust Multichip Average (RMA) using Affymetrix Power Tools (version 1.12.0). Probe sets that had below-median levels of expression in all arrays were removed. Differential expression was determined using linear modelling of the time-course using the Limma package (version 3.20.0)[95]. All $P$-values are corrected for multiple testing;

using a method derived from Benjamini and Hochberg's method to control the false-discovery rate[96].

Transcripts were clustered based on their expression patterns over the time-course using a soft-clustering approach (MFUZZ)[97]. Gene ontology was determined by the HOMER (Hypergeometric Optimisation of Motif EnRichment, version 4.7) programme[98]. Fold-change per gene ontology term was determined by: (number of target genes in term / total number of target genes) / (total number of genes in term / total number of genes in background list).

Temporal variation in gene expression was assessed by fitting a temporal trend using a regression spline with 3 df (Limma, 3.22.7). $P$-values were adjusted for multiple testing, controlling the false-discovery rate (FDR) below 1%. TC annotations were compiled from NetAffx (access date 30/06/2020) and hugene21sttranscript-cluster.db (8.5.0). Common temporal expression patterns were sought amongst differentially expressed genes using the unsupervised classification technique Mfuzz (2.26.0), informed by the minimum distance between cluster centroids (Dmin).

**Network analysis**. Co-abundance networks were constructed using Weighted correlation network analysis (WGCNA)[99]. We constructed WGCNA co-abundance networks separately using the CELF ptOP, CELF LLL and BUS ptOP samples, including any OTUs that appeared in 20% of samples in at least one of these four subsets (646 OTUs). Spearman correlation was used to construct the WGCNA adjacency matrices. OTU reads were transformed using $\log(x + 1)$ prior to WGCNA analysis.

**Statistics and reproducibility**. Statistical tests and their interpretation are described in the context of individual methods above.

**Reporting summary**. Further information on research design is available in the Nature Portfolio Reporting Summary linked to this article.

## Data availability
Raw sequence data for the bacterial isolates have been deposited in the European Nucleotide Archive at the European Bioinformatics Institute under accession number ERP110629. Assembled genomes are available with the number PRJNA578828. The raw OTU data for the Celtic Fire study is available with the accession number PRJEB40753, and that from the Busselton study with the accession number PRJEB29091. The gene expression data for airway epithelial differentiation is deposited at the EGA Archive with the ID: EGAS00001006689. The source data for Fig. 3a, e, g is available at https://figshare.com/articles/dataset/Genomic_attributes_of_airway_commensal_bacteria_2023/24901788. Source data for Supplementary Fig. 8 is available at https://figshare.com/articles/dataset/GENOMIC_ATTRIBUTES_OF_AIRWAY_COMMENSAL_BACTERIA_AND_MUCOSA-_Supplementary_figure_8/24983193.

## Code availability
All data analysis scripts are available online at https://zenodo.org/records/10466935 (reference[100]).

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

## Acknowledgements

The culture collection was funded primarily by the Asmarley Trust. Isolate sequencing was funded by the Wellcome Trust (WT098051; WT206194 and 108413/A/15/D), and we thank the Wellcome Sanger Institute Pathogen Informatics and Research Support Facility for supporting this research. Jonathan Ish-Horowicz was the recipient of a Wellcome Trust PhD studentship (215359/Z/19/Z). Bioinformatic investigation of isolated genomic sequences was supported by MDC Berlin DFG SFB1449: "Dynamic Hydrogels"; KFO339; "FOOD@"; DFG SFB1365: "Renoprotection"; and JPI-AMR: EMBARK. Genomic studies of airway transcripts were supported by a joint Wellcome Senior Investigator Award to WOCC and MFM (WT096964MA and WT097117MA). The Busselton Healthy Ageing Study is funded by grants from the Government of Western Australia (Office of Science, Department of Health) and the City of Busselton, and from private donations to the Busselton Population Medical Research Institute. We thank the WA Country Health Service and the community of Busselton for their ongoing support and participation.

## Author contributions

M.F.M. and W.O.C. planned the overall study structures; T.D.L. suggested building a culture and sequence collection of airway bacteria, and led sequencing at the Wellcome Sanger Centre; L.C., C.C., M.C., and M.F.M. designed and carried out the microbial culture of airway samples; C.C. has catalogued and biobanked the organisms; S.K.F. led bioinformatic strategy for microbial sequences, which were carried out by U.L., J.I.-H., Th. U.P.B. and Ti. B. with advice from S.K.F. and S.F.; M.T.O. carried out analyses of metabolomic data, with guidance by M.D.; C.M.B. carried out the microbial community analyses from the Celtic Fire Study with input from C.C., J.I.-H. and L.C.; C.B., O.O.'C., J.F., G.D., K.L., J.C.-T., M.A., J.M.H., R.G., and K.F.C. designed and completed clinical and bronchoscopic investigations of patients and volunteers in the Celtic Fire Study; S.D. and A.M. co-ordinated clinical data and sample collection and N.K. managed isolate sequencing; J.P. designed and completed the time-series analysis of gene expression and metabolite production during airway epithelial differentiation, with bioinformatic analysis from S.P. and S.W.O.; E.T. performed microbial community analyses from the Busselton Survey, with contributions from J.I.-H., L.C. and M.J.C.; the Survey itself was led by A.W.M., J.H., M.H., and A.J. W.O.C.M. co-ordinated the first draft of the paper, but all authors contributed to the writing and revision.

## Competing interests

The authors declare no competing interests.
