## [Peer Review File · Communications Biology]

Reviewers' comments:

Reviewer #1 (Remarks to the Author):

Cuthbertson et al. have characterized airway microbiome and mucosa. They performed whole-genome sequencing of the principal airway bacterial species. Using whole genome content, they identify dysbiotic features that can presage homeostatic breakdown during acute attacks of asthma and chronic obstructive pulmonary disease. In addition, they match the gene content of isolates to host transcripts and metabolites expressed late in airway epithelial differentiation. This paper is interesting, but there are several problems. Some issues need to be addressed. Please see the comments below.

1. The letters on most figures are too small to read. And insufficient resolution.
2. Although the authors used NHBE isolated from a 26-year-old adult, is it possible to confirm similar results with other donors' NHBE?
3. The TEER value of NHBE-ALI should be introduced in the main manuscript.
4. The results of the ELISA analysis of MUC5AC in NHBE-ALI should be introduced in the main manual.
5. Results of microarray analysis have to be deposited in GEO (or other databases).
6. Is it possible to co-culture airway microbiome and NHBE-ALI?

Reviewer #2 (Remarks to the Author):

The study is an in-depth catalogue of the bacteria present in airway brushes from several healthy patients and patients with asthma. The authors have done an enormous amount of work to culture and sequence lung bacteria and to investigate their genetic potential. They then examine the oral and lung microbiome in a larger group of healthy and asthmatic patients that included multiple cohorts. Finally, they perform metabolomics analyses and gene expression analyses of cultured lung epithelial cells and relate these to the microbial communities. The cataloguing of the bacteria and genes in the airway is interesting, but the inferences that can be drawn about the function and impact of these bacteria and genes in health and disease is limited.

>The abstract and introduction overstate and simplify the state of knowledge of the lung microbiome and should be revised. We do not have certainty around the composition, dynamics, and function of the normal lung microbiome or its role in various lung diseases. The conclusion that they identified dysbiotic features that predict acute attacks of asthma and COPD are not supported by the study design. Statements such as "pulmonary diseases arise in the intrathoracic airways" are rather vague and inaccurate.

>The patient characteristics beyond asthma and control do not seem accounted for? The DMM modeling is difficult to interpret given the inclusion of different sample sites and multiple samples from the same individual.

>Are there sequencing and bronchoscopy controls?

>It is unclear how to relate the airway epithelial cell culture results to the microbial findings given that the epithelial cells are from one patient and may have different properties in cell culture.

>While the manuscripts contains a large amount of work and data, the link to any disease process is

unclear.

Reviewer #3 (Remarks to the Author):

Comments for EXTENSIVE NOVEL CAPACITIES OF THE AIRWAY MICROBIOME AND MUCOSA

In this study, the authors present a fundamental yet systematic characterization of the predominant airway bacterial species, drawing upon both culture-dependent and whole-genome sequencing methodologies. A total of 52 novel species were identified among 126 organisms, which together accounted for 75% of the commensal organism abundance. To investigate the genetic attributes of these species, the authors conducted functional characterization, evolutionary analysis, and comparisons with amplicon sequencing results from representative human samples. The comprehensive analysis holds clinical relevance for a potential deeper understanding of asthma and chronic obstructive pulmonary disease. However, certain aspects of the study warrant further revision, as detailed below:

1. The annotation process for the sequenced genomes appears ambiguous. While the NCBI and GTDB systems differ, it is unclear whether the authors employed either or both systems in their analysis. Please provide a rationale for the chosen methodology in this study.
2. The authors filtered draft genomes without considering standard quality control measures. Did the authors utilize widely accepted criteria, such as MISAG, to exclude low-quality genomes prior to downstream analysis?
3. As this study aims to provide a foundation for further investigation into the microbial community, the inclusion of viruses and fungi alongside bacteria is crucial. Please elaborate on this aspect and provide information on the proportion of these microorganisms within the total microbial community. If the authors focused solely on bacteria in their 16S analysis, it would be more accurate to state that the novel species constitute 75% of bacterial diversity.
4. In the analysis of antimicrobial resistance genes, the authors employed AMRfinder and ARIBA tools, as well as the CARD database. It would be beneficial to provide additional information about the identified resistance genes, including whether they arise from point mutations, stress response genes, or other factors, in order to enhance the study's comprehensiveness.
5. To what extent do these 52 novel species contribute to the existing knowledge of lower respiratory commensal bacteria in the literature? Please provide further discussion and relevant references.
6. The logic in lines 383-387 could be clarified. Firstly, the sentence "Our results will greatly improve metagenome assembly and allow assays of individual microbial activities through metatranscriptomics" is difficult to follow. Secondly, it is important to note that metagenomic sequencing should be sufficiently deep to capture less abundant microbes, but not for those that are already abundant. Lastly, it is unclear whether the authors emphasize metagenomics to suggest a lack of assembly applications in previous studies focusing on lower respiratory commensal isolates.
7. The manuscript could benefit from improved organization and a more concise presentation. For instance, the subtitle in line 62 is unnecessary within the introduction section. Additionally, lines 282-283 and 289-295 could be combined into other paragraphs in the results section.

Point-by-point rebuttal

Referee #1: Lung organoids and viral infection models

General

Cuthbertson et al. have characterized airway microbiome and mucosa. They performed whole-genome sequencing of the principal airway bacterial species. Using whole genome content, they identify dysbiotic features that can presage homeostatic breakdown during acute attacks of asthma and chronic obstructive pulmonary disease. In addition, they match the gene content of isolates to host transcripts and metabolites expressed late in airway epithelial differentiation. This paper is interesting, but there are several problems. Some issues need to be addressed. Please see the comments below.

1.0. The letters on most figures are too small to read. And insufficient resolution.

We have included higher-resolution versions of the figures in the resubmission. We have very high-resolution images that we will upload if the paper is accepted for publication.

1.1. Although the authors used NHBE isolated from a 26-year-old adult, is it possible to confirm similar results with other donors' NHBE?

The reviewer raises the important point of host variation in responses to bacteria. These may be genetic or epigenetic, and multiple factors are likely to be at play.

We now acknowledge this on line 427-431 in the Discussion, "We have studied HAEC from a single donor, and it is to be expected that multiple genetic and epigenetic factors will influence different components of the pathways we have identified. Such factors may be systematically investigated by knockdown and knock-in in model systems and by culture of HAEC from subjects with and without airway diseases⁷⁰".

Although for space not included in the revision, our direct experience is that genome wide association studies of asthma have identified approximately 100 loci that modify asthma risk, half of which are expressed in the epithelium, and many of which have a likely role in infection. The Odds Ratios of these loci are $\ll 1.5$, and the sample size to detect them has been tens of thousands of asthmatics and hundreds of thousands of controls. Additionally, GWAS studies of influences on the abundance of microbial taxa in the bowel have failed to identify convincing genetic effects. In these circumstances, testing additional cell lines may be less rigorous than it appears.

1.2. The TEER value of NHBE-ALI should be introduced in the main manuscript.

We now state on lines 315-317 that "HAEC from a single donor were grown in triplicate and harvested on days 0, 2, 3, 7, 14, 21 and 28. Trans-epithelial resistance (TEER) rose from 7.4 ± 0.3 at day 0 to 1551 ± 113 on day 28, and MUC5AC mRNA production rose 30-fold over the same period (Supplementary Figure 4), indicating full epithelial development".

1.3. The results of the ELISA analysis of MUC5AC in NHBE-ALI should be introduced in the main manual.

See 1.2 above.

1.4. Results of microarray analysis have to be deposited in GEO (or other databases).

On lines 968-969 under DATA AVAILABILITY, we now state “The gene expression data for airway epithelial differentiation is deposited at the EGA Archive with the ID: EGAS00001006689.”

1.5. Is it possible to co-culture airway microbiome and NHBE-ALI?

It is possible to co-culture, but beyond the remit of the study. We have now extended the Discussion in lines 399-401 to state “Inhibition of inflammation in airway epithelial cell models has recently been shown for *Rothia*, *Prevotella* and *Streptococcus* spp. grown from children with CF^{69,70}”.

We reiterate this point when discussing the limitation of using a single cell donor in lines 430-431, “It is already clear that co-culture of pathogens and commensals in such models will reveal many further pathways underpinning host-microbial interactions^{69,70}.”

Referee #2: Microbiome and metabolome in lung disease

General

The study is an in-depth catalogue of the bacteria present in airway brushes from several healthy patients and patients with asthma. The authors have done an enormous amount of work to culture and sequence lung bacteria and to investigate their genetic potential. They then examine the oral and lung microbiome in a larger group of healthy and asthmatic patients that included multiple cohorts. Finally, they perform metabolomics analyses and gene expression analyses of cultured lung epithelial cells and relate these to the microbial communities. The cataloguing of the bacteria and genes in the airway is interesting, but the inferences that can be drawn about the function and impact of these bacteria and genes in health and disease is limited.

2.1. The abstract and introduction overstate and simplify the state of knowledge of the lung microbiome and should be revised. We do not have certainty around the composition, dynamics, and function of the normal lung microbiome or its role in various lung diseases. The conclusion that they identified dysbiotic features that predict acute attacks of asthma and COPD are not supported by the study design.

We agree with the Reviewer. Although we hoped to draw attention to the potential for understanding the complexity by identifying structures such as correlation networks, pulmotypes and potential host factors, we accept that we didn't differentiate adequately between the actual state of knowledge and conjecture about the future.

We have therefore changed the text to make clear that we have not identified dysbiotic features that predict acute attacks of asthma and COPD. In the Abstract in lines 53-54 we now write, “Using whole-genome content we identify dysbiotic features that may influence asthma and chronic obstructive pulmonary disease “; and in the Discussion lines 434 -437 we now state, “Here we have found networks of interacting bacteria that are attenuated in the lower airways, possibly presaging loss of stability⁷⁹. The hypothesis can now be tested that airway dysbiosis and microbial community instability predisposes to dysregulation of inflammatory processes during acute exacerbations of lung disease”.

2.2. Statements such as “pulmonary diseases arise in the intrathoracic airways” are rather vague and inaccurate.

We now state on lines 72 to 74 in the introduction that “Common pulmonary diseases including asthma, COPD, bronchopneumonia, cystic fibrosis and lung cancer arise in the intrathoracic airways, whose commensal microbiota are similar to those of the oropharynx^{13,21,22}.”

2.3. *The patient characteristics beyond asthma and control do not seem accounted for? The DMM modelling is difficult to interpret given the inclusion of different sample sites and multiple samples from the same individual.*

We have carried out additional analyses and extensively re-written the section on Dysbiosis. An advantage of the DMM analyses is that the models specifically take into account the effects of different sample sites and multiple samples from the same subjects.

We now state on lines 279-305 that, “We explored underlying components of airway communities by using Dirichlet Multinomial Mixtures (DMM)⁵⁴ on all samples from the BUS and CELF subjects, finding that samples formed predominantly into two clusters (Airway Community Type 1 and 2: ACT 1 and ACT2) (Figure 3a). The main drivers for the two pulmotype clusters were identified as *Streptococcus*, *Veillonella*, *Prevotella* and *Haemophilus* spp. in descending order of relative abundance across all samples. ACT1 was dominated by *Streptococcus*, *Veillonella* and *Prevotella* in 410 samples; whilst ACT2 was dominated by *Streptococcus*, *Veillonella* and *Haemophilus* in 478 samples (Figure 3a). Principal coordinates analysis based on Bray-Curtis-distance (β -diversity) of the airway microbiota confirmed significant overall compositional differences between the two community type clusters (PERMANOVA P -value > 0.001) (Figure 3b).

Congruence analysis of CELF samples (Figure 3c) confirmed consistency in assignment for samples coming from the same donor ($\chi^2 < 0.005$) or the same sampling site ($\chi^2 < 0.005$).

We performed univariate analysis to investigate the association between CELF subject metadata and potential indicators of dysbiosis, specifically, evenness and richness (Figure 3d), and bacterial abundance at phylum level (Figure 3e). Features describing clinical phenotypes and sample origin were often strongly collinear. We therefore assessed found associations in turn for retained significance with each potential confounder, using a nested rank-transformed mixed model test⁵⁵ and considering repeated sampling of patients as a random effect.

We saw pervasive effects both on alpha diversity and phylum level of the tested predictors (Figure 3d and 3e). Importantly, the Shannon index and richness were significantly decreased with asthma status and severity (MWU false discovery rate (FDR) < 0.1) (Figure 3d).

We found an increase (although not significantly) of the *Proteobacteria* Phylum associated with asthma status (Figure 3e), in line with the taxonomic profile of patients with asthma vs. healthy controls (Figure 3g). This is consistent with many reports of *Proteobacteria* excess in asthmatic airways^{8,9,56}.

We examined the impact of study, asthma status, and sampling site on the distribution of community types in the CELF thoracic samples, using logistic regression models with sex and age as control variables. The results indicated significant differences in ACT proportions across different sampling sites: LUL vs OTS: Odds Ratio 95% confidence interval 0.135 - 0.444

(p-val: 3.1e-07); LLL vs. OTS: 0.049 - 0.249 (p-val: 5.0e-10). Statistical significance was more marked for the left upper lobe (FDR q-value <0.001) than the left lower lobe (q<0.10). “

2.4 Are there sequencing and bronchoscopy controls?

We now add in the Methods on line 578 “Scope control washes were taken at each bronchoscopy” and on lines 593-596 “Within a Class 2 biological safety cabinet, each bronchial brushing was transferred directly into an LME tube. To control for contamination an empty LME tube (i.e., an extraction control) was added to each batch. The extraction control underwent the entire extraction process along with the samples. Eighteen two randomly selected Scope Control Washes (SCWs) also underwent DNA extraction.

As already stated on lines 615-617 “Contaminant OTUs were identified using Spearman’s correlation between bacterial biomass with number of reads per samples. OTUs were considered to be contaminants with a Benjamini-Hochberg corrected *P*-value of <0.05 and a correlation value of >0.2.”

2.5. It is unclear how to relate the airway epithelial cell culture results to the microbial findings given that the epithelial cells are from one patient and may have different properties in cell culture.

We have addressed this in the response to Reviewer 1 in points 1.1 and 1.5 above. We acknowledge in the text that there will be differences in responses that will need to be considered in co-culture.

2.6 While the manuscripts contains a large amount of work and data, the link to any disease process is unclear.

As we write in the Introduction, the purpose of our investigations has been to provide reference data for future studies of the airway microbiome and mucosa. We have discovered a high degree of novelty in organisms, their genomic content, and their community structure.

Referee #3: Microbial metagenomics

General

In this study, the authors present a fundamental yet systematic characterization of the predominant airway bacterial species, drawing upon both culture-dependent and whole-genome sequencing methodologies. A total of 52 novel species were identified among 126 organisms, which together accounted for 75% of the commensal organism abundance. To investigate the genetic attributes of these species, the authors conducted functional characterization, evolutionary analysis, and comparisons with amplicon sequencing results from representative human samples. The comprehensive analysis holds clinical relevance for a potential deeper understanding of asthma and chronic obstructive pulmonary disease.

We are grateful for these encouraging comments.

3.1. The annotation process for the sequenced genomes appears ambiguous. While the NCBI and GTDB systems differ, it is unclear whether the authors employed either or both systems in their analysis. Please provide a rationale for the chosen methodology in this study.

We thank the reviewer for pointing this aspect out. We used the MIGA annotation with the NCBI database to annotate the genomes and evaluate the likelihood of novel species detected. To test the robustness of the annotation, we further annotated the genomes with

another of the gold standards, GTDB. We carefully compared the results of both methods as we are aware that the NCBI database as well as GTDB have some flaws. We therefore reported three categories, which are: species which are fully supported by both methods; species which are only supported by one of the two methods; and putatively novel species which could not be assigned by any of the databases.

3.2. The authors filtered draft genomes without considering standard quality control measures. Did the authors utilize widely accepted criteria, such as MISAG, to exclude low-quality genomes prior to downstream analysis?

We checked the quality of the assembled genomes using BACTOPIA and removed potentially contaminated isolates. Within the MIGA pipeline, the genomes were checked for 106 essential bacterial genes. We have added information about the genome quality as well as the genome completeness in Supplementary Tables S1 and S2. By these measures, the MISAG criteria reporting assembly quality, genome completeness and measure of contamination should be fulfilled.

We now state in lines 99 to 104, “The Bactopia quality report for the genome assemblies is reported in Supplementary Table 1. Forty-four isolates were annotated to species level in accordance with MIGA²⁶ (TypeMat and NCBIProk) and with GTDBtk. A further 30 species were identified by either MIGA (TypeMat and NCBIProk) or GTDBtk. The genome completeness and the contamination percentage was tested within the MIGA pipeline aligning 106 bacterial core genes²⁷ (Supplementary Table 2)”.

3.3. As this study aims to provide a foundation for further investigation into the microbial community, the inclusion of viruses and fungi alongside bacteria is crucial. Please elaborate on this aspect and provide information on the proportion of these microorganisms within the total microbial community. If the authors focused solely on bacteria in their 16S analysis, it would be more accurate to state that the novel species constitute 75% of bacterial diversity.

The focus of our study has been to investigate the bacterial microbiome. We now make this explicit in the Title and throughout the text.

Respiratory tract viruses have been the subject of numerous investigations and are well understood, for which we now give key references in lines 77-78 in the Introduction.

A fungal presence is recognised in the oropharynx, and in the lower airways fungi play an important part in chronic infective disease such as Cystic Fibrosis and Bronchiectasis. However, we were unable to detect fungi in the intrathoracic airways of our subjects either by ITS2 amplification or by culture.

We therefore state on lines 77-84, “The airway microbiota encompass viruses, fungi and bacteria²⁰. A variable viral microbiome (excluding phage) is well described at the molecular level^{20,21}. Oropharyngeal fungi such as *Candida* and *Aspergillus* spp. are commonly cultured from asthmatics, confounded by therapy with inhaled corticosteroids. Although important in cystic fibrosis and bronchiectasis²², fungi have very low biomass in the lower airways of healthy individuals²³. Airway commensal bacteria from healthy subjects have not previously been systematically cultured or sequenced. This lack has limited the structured study of interactions between bacteria, viruses, fungi and mucosal immunity in clinical samples or in model systems. In this paper we describe such systematic exploration, substantially extending what is known about core constituents of airway bacterial communities.”

3.4. In the analysis of antimicrobial resistance genes, the authors employed AMRfinder and ARIBA tools, as well as the CARD database. It would be beneficial to provide additional information about the identified resistance genes, including whether they arise from point mutations, stress response genes, or other factors, in order to enhance the study's comprehensiveness.

All information generated by AMRfinder and ARIBA run within the BACTOPIA pipeline has now been added to the manuscript as Supplementary Table S5. This involves detailed description of each hit, including separate designation of allelic (e.g. protective point mutations) vs genic (more conventional ARGs) resistance elements and the designation for each. We are confident that with this presentation, the reader can perform further filtering and sub-setting within the results to follow up with questions such as those the reviewer outlines.

3.5. To what extent do these 52 novel species contribute to the existing knowledge of lower respiratory commensal bacteria in the literature? Please provide further discussion and relevant references.

We now expand on this important topic in lines 390-411 of the discussion: “Our results describe the first systematic culture, isolation and sequencing of the respiratory commensal bacteria. Although the principal airway phyla are well known through OTU studies of whole communities, previous attempts at culture have been limited to patients with Cystic Fibrosis (CF)⁷²⁻⁷⁴, a disease in which CFTR mutations induce major changes in the airway mucosal fluid and host environment. Anaerobic species cultured from these studies include the genera *Actinomyces*, *Atopobium*, *Micrococcus*, *Neisseria*, *Prevotella*, *Rothia*, *Streptococcus*, and *Veillonella*⁷³, and may be similar to our isolates. Nevertheless, systematic commensal sequencing has not previously been carried out, and 40% of our isolates are novel species. Their gene content indicates a wide range of previously undocumented capacities to interact with other organisms and the airway mucosa.

Streptococcus genera showed the greatest novelty, with 60% of isolates not previously found in public databases. These are in phylogenetic clusters distinct from known respiratory commensals such as *S. salivarius* and *S. parasanguinis*. Their abundance in the oropharynx and lower airways suggest important functions that are yet to be explored.

Our findings mean that it is now possible to investigate systematically the effects of individual bacteria and their combinations on airway inflammation and infection. Therapies derived from healthy microbial communities are established for inflammatory and metabolic bowel diseases, through faecal transplantation, bacteriotherapy with specific organisms⁷⁵, and bacterial metabolites⁷⁶. Inhibition of inflammation in airway epithelial cell models has recently been shown for *Rothia*, *Prevotella* and *Streptococcus* spp. grown from children with CF^{73,74}.

Rich microbial environments are well known to protect against asthma in schoolchildren⁷⁷ and adults⁷⁸, although the responsible organisms have not been identified in airway communities. We have found previously shown *Selenomonas*, *Megasphaera* and *Capnocytophaga* spp., to be reduced in abundance in asthmatic ptOP samples⁵, but despite their moderate abundance ((0.4-2.8% of the total) we have not managed to culture them. Future isolation is desirable to test if they are indicator species or direct contributors to respiratory health.”

3.6. The logic in lines 383-387 could be clarified. Firstly, the sentence "Our results will greatly improve metagenome assembly and allow assays of individual microbial activities through metatranscriptomics" is difficult to follow. Secondly, it is important to note that metagenomic sequencing should be sufficiently deep to capture less abundant microbes, but not for those that are already abundant. Lastly, it is unclear whether the authors emphasize metagenomics to suggest a lack of assembly applications in previous studies focusing on lower respiratory commensal isolates.

We have clarified this point as follows in lines 422-432: "Metagenomic and metatranscriptomic sequencing has been very informative in understanding bowel microbial activities in health and disease. In contrast, non-purulent airway secretions typically contain <5% microbial DNA⁸³ and are difficult to access. Purulent secretions, such as sputum, are often heavily contaminated with upper airway and oral flora²⁴. Consequently, metagenomic sequencing of respiratory samples has so far identified the most abundant pathogens and commensals, with limited functional resolution^{24,83,84}. By extending available airway genome and gene catalogue data as we have here, sequenced reads too sparse to reliably assemble per sample can be mapped to our gene and genome assemblies. This will provide a scaffold for metagenome analyses as well as for the selection of marker genes and primers adapted for targeted amplicon sequencing of specific airway microbiota. As we have shown above, gene content of airway communities can also be inferred by mapping genome sequences to OTU results. Thus, through the present collection, taxonomic and functional characterization of airway communities is facilitated."

3.7. The manuscript could benefit from improved organization and a more concise presentation. For instance, the subtitle in line 62 is unnecessary within the introduction section. Additionally, lines 282-283 and 289-295 could be combined into other paragraphs in the results section.

We have revised the organisation of the paper throughout, as suggested.

REVIEWERS' COMMENTS:

Reviewer #1 (Remarks to the Author):

The authors have perfectly addressed all my comments.
This paper will bring useful information to many readers.
I think no additional revision required.

Reviewer #2 (Remarks to the Author):

The authors have responded to many of the reviews. Prior critiques had asked for revision of statements regarding what is known about the role of the lung microbiome. This has been done somewhat, but there are still overstatements that need to be modified. Recommend revising the first 2 sentences of the abstract as these are not known facts in the field. The sentence in the abstract about lupus should also be removed as not supported by data in the paper.

In the introduction, would revise the sentence that asthma and COPD are "driven by infections." This is an overstatement and suggests that infections are the main cause of these diseases, which is not the case. Would also revise: "Microbial community dysbiosis with overgrowth of pathobionts underlies asthma, COPD, pneumonia, and other pulmonary disorders." to not imply that microbes are the underlying cause of these diseases.

The link of gene expression in HAECs from a single donor to the microbiome are not direct, and not sure it adds much to the manuscript especially given the increasing numbers of single cell data sets available in the lung.

Reviewer #4 (Remarks to the Author):

Cuthbertson et al. have addressed the comments raised by Reviewer #3. However, I have a couple of additional comments for the authors, which they can use their discretion to address.

1. Define "new species": Please provide a more detailed description of how a "new species" is defined in your study. This will help readers understand the criteria used to identify and classify new species among the bacterial isolates.

2. Rarefaction plot: While you have reported 126 commensal bacterial species, it is likely that this list is not exhaustive. Please clarify how many bacterial colonies were sequenced in total and provide a rarefaction plot. This plot will visualize the relationship between the number of colonies sequenced and the number of species detected. It will allow readers to assess the likelihood of identifying additional species with further culture and sequencing efforts.

Reviewer #1

*The authors have perfectly addressed all my comments.
This paper will bring useful information to many readers.
I think no additional revision required.*

Thank you!

Reviewer #2

The authors have responded to many of the reviews. Prior critiques had asked for revision of statements regarding what is known about the role of the lung microbiome. This has been done somewhat, but there are still overstatements that need to be modified. Recommend revising the first 2 sentences of the abstract as these are not known facts in the field. The sentence in the abstract about lupus should also be removed as not supported by data in the paper.

We have revised the first sentence of the abstract (line 46) to add “**in likelihood** reflecting co-evolution with human host factors”, as co-evolution between host and microbiota is accepted as generally true for other microbiomes.

We stand by the statement “the airway microbiome underpins cognate management of mucosal immunity and pathogen resistance” as this is exemplified in reference 9. We have removed the sentence about lupus.

In the introduction, would revise the sentence that asthma and COPD are “driven by infections.” This is an overstatement and suggests that infections are the main cause of these diseases, which is not the case.

We have modified the statement (line 62) to read more accurately “and **acute exacerbations of both diseases** are driven by respiratory infections”. Exacerbations of asthma and COPD fell by 50% internationally following non-pharmaceutical interventions to restrict COVID transmission and the role of infection is not controversial.

Would also revise: “Microbial community dysbiosis with overgrowth of pathobionts underlies asthma, COPD, pneumonia, and other pulmonary disorders.” to not imply that microbes are the underlying cause of these diseases.

We have modified the sentence on line 75 to read “**accompanies**” rather than “underlies”.

The link of gene expression in HAECs from a single donor to the microbiome are not direct, and not sure it adds much to the manuscript especially given the increasing numbers of single cell data sets available in the lung.

We understand that this is a philosophical point.

We already address the single donor issue in the Discussion lines 431-434 “We have studied HAEC from a single donor, and it is to be expected that multiple genetic and epigenetic factors will influence different components of the pathways we have identified. Such factors

may in future be systematically investigated by knockdown and knock-in in model systems and by culture of HAEC from subjects with and without airway diseases”.

We have examined the single cell data from airway epithelial cells in the Protein Atlas: this does not provide helpful information for most of the genes and pathways identified in our study.

Context to global gene expression data may be derived spatially, or dynamically or through co-expression network analyses. Each modality gives different information.

We studied the dynamic changes in expression that occur during epithelial differentiation, with the hypothesis on lines 316-8 “that the transition from monolayer to ciliated epithelium over 28 days would be accompanied by progressive expression of genes and secretion of metabolites for managing the microbiota”.

Reviewer #4

Cuthbertson et al. have addressed the comments raised by Reviewer #3. However, I have a couple of additional comments for the authors, which they can use their discretion to address.

1. Define "new species": Please provide a more detailed description of how a "new species" is defined in your study. This will help readers understand the criteria used to identify and classify new species among the bacterial isolates.

We now expand on new species in lines 108-111 “We defined a 'new species' when isolates could not be assigned to known species in reference databases. We classified isolates as 'putatively novel species' when they exhibited no close relation to any species in the TypeMat or NCBI Prokaryotic Databases, determined by the MIGA tool with a p-value threshold of 0.05 and an incongruent species assignment indicated by gtdbtk”.

2. Rarefaction plot: While you have reported 126 commensal bacterial species, it is likely that this list is not exhaustive. Please clarify how many bacterial colonies were sequenced in total and provide a rarefaction plot. This plot will visualize the relationship between the number of colonies sequenced and the number of species detected. It will allow readers to assess the likelihood of identifying additional species with further culture and sequencing efforts.

We have constructed a rarefaction plot as suggested, but the data is sparse (cultures taken on five occasions) and it is difficult to interpret the curves with confidence. We think it less elegant but simpler to give the actual numbers in lines 96-98 “We cultured 651 isolates, 256 of which were successfully whole-genome sequenced. Of these, five sequences appeared mixed and were excluded. After removing duplicates on a 99.5% nucleotide identity threshold, 126 unique strains remained”.

We address the extent to which cultured organisms are representative in the section “Community Coverage” from lines 226-253. Here we have identified important gaps in coverage by comparison with 16S data from multiple individuals in different populations.